# Towards Self-Refinement of Vision-Language Models with Triangular Consistency

**Yunlong Deng**[1*], **Guangyi Chen**[1,2*], **Tianpei Gu**[3], **Lingjing Kong**[2], **Yan Li**[1], **Zeyu Tang**[2], and **Kun Zhang**[1,2]

[1]Mohamed bin Zayed University of Artificial Intelligence
[2]Carnegie Mellon University
[3]ByteDance US

## Abstract

Vision-Language Models (VLMs) integrate visual knowledge with the analytical capabilities of Large Language Models (LLMs) through supervised visual instruction tuning, using image-question-answer triplets. However, the potential of VLMs trained without supervised instruction remains largely unexplored. This study validates that VLMs possess inherent self-refinement capabilities, enabling them to generate high-quality supervised data without external inputs and thereby learn autonomously. Specifically, to stimulate the self-refinement ability of VLMs, we propose a self-refinement framework based on a Triangular Consistency principle: within the image-query-answer triangle, any masked elements should be consistently and accurately reconstructed. The framework involves three steps: (1) We enable the instruction generation ability of VLMs by adding multi-task instruction tuning like image→question-answer or image-answer→question. (2) We generate image-query-answer triplets from unlabeled images and use the Triangular Consistency principle for filtering. (3) The model is further updated using the filtered synthetic data. To investigate the underlying mechanisms behind this self-refinement capability, we conduct a theoretical analysis from a causal perspective. Using the widely recognized LLaVA-1.5 as our baseline, our experiments reveal that the model can autonomously achieve consistent, though deliberately modest, improvements across multiple benchmarks without any external supervision, such as human annotations or environmental feedback. We expect that the insights of this study on the self-refinement ability of VLMs can inspire future research on the learning mechanism of VLMs. Code is available at SRF-LLaVA.

## 1 Introduction

Recent advancements in Large Language Models (LLMs) have demonstrated remarkable capabilities in natural language understanding and generation. By incorporating visual features into the linguistic modalities understandable by Large Language Models (LLMs), Vision-Language Models (VLMs) have developed the capability to interpret visual content effectively. A key factor in the success of these VLMs [1, 2, 3, 4, 5, 6, 7] is visual instruction tuning, which aligns visual content with the representation of LLM by optimizing the model's responses to visual instructions.

Acquiring supervised data with high-quality annotations is crucial for the success of visual instruction tuning. However, compiling visual instructions, such as image-question-answer triplets, is non-trivial. Such data cannot be directly crawled from the web and requires human involvement, making the process both challenging and costly. Careless data collection may lead to copyright issues. These factors motivate researchers to leverage synthetic data instead.

39th Conference on Neural Information Processing Systems (NeurIPS 2025).

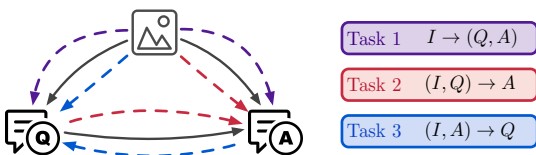

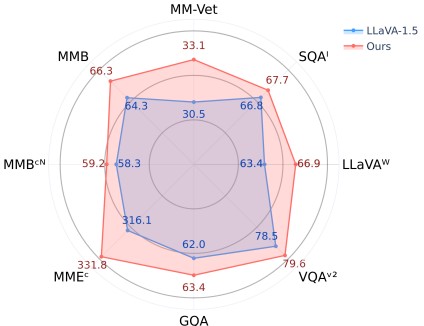

(a) **The illustration of the Triangular Consistency principle.** This principle evaluates the consistency within image-question-answer triplets. Specifically, we independently mask A, Q, or both, and use VLMs to infer the masked elements. We then check consistency by assessing whether the generated outputs align with the original ones.

(b) **Stimulating self-improvement boosts downstream performance.**

The key to generating synthetic data lies in the creation of question-answer instructions. To ensure high-quality supervision signals, several methods [8, 9, 10, 11] utilize advanced VLMs, such as GPT-4V [12] and Gemini [13], to generate captions for unlabeled images. For example, ShareGPT4V [8] applies GPT-4V to generate a curated set of 100K high-quality captions and expands the dataset to 1.2M using a captioner trained on these captions. Additionally, some methods [14, 15, 16] leverage advanced LLMs, such as GPT-4 [17], as expert evaluators of generated data quality. However, these approaches depend on high-quality VLMs, which may face bottlenecks due to usage limits or the cost of proprietary models. As models improve, it becomes increasingly challenging to find superior teacher models for data annotation. To address this issue, some methods propose deriving annotation signals from environmental feedback, such as evaluation performance [18, 19, 20, 21]. However, relying on environmental feedback for supervision signals may be inefficient for model training. **More discussion of the related work can be found in Appendix A.**

Motivated by the challenges outlined above, this paper seeks to address the following question:

*Can we refine VLMs without relying on external supervision, using only the model itself?*

To stimulate the self-refinement capabilities of VLMs, we propose a framework using a Triangular Consistency principle to generate high-quality instructions by the model itself. This framework consists of three steps. (1) We fine-tune the VLMs to enhance their ability to generate instructions through a multi-task objective. This involves randomly masking the question, answer, or both, and training the VLMs to reconstruct the missing components. Consequently, the models can generate query-answer pairs from unlabeled images. (2) To ensure high-quality generated instructions, as shown in Figure 1a, we filter them with Triangular Consistency by inferring one component (question or answer) based on the other, and then compare the consistency between new-inferred ones and the original parts in the instructions. By filtering, we select the instructions with high consistency. (3) Finally, we leverage the selected instructions to refine the VLMs. This structured framework enhances the model's capability through continuous improvement, using its outputs as a feedback mechanism. Consequently, it supports multiple iterations of enhancement using only unlabeled images.

To investigate the underlying mechanisms behind the self-refinement capability of VLMs, we propose both theoretical and experimental analyses for validation. Theoretically, we formulate the refinement process as a semi-supervised learning task. Initially, we identify the causal relations between natural language and images. Subsequently, we offer guarantees that learning can be effectively conducted using only unlabeled images, grounded in the principles of causality. Then, experimentally, we employ LLaVA-1.5 [2] as our baseline, and evaluate whether the self-generated instructions can refine the models. As demonstrated by the results across multiple benchmarks, which are shown in Figure 1b, the proposed framework can efficiently stimulate the self-refinement ability of VLMs. Our primary contributions are summarized as follows:

- **Triangular Consistency Principle**: We introduced the Triangular Consistency principle as a measure for testing the reliability of generated instructions on unlabeled data.
- **Self-Refinement Framework**: We developed a self-refinement framework to update the model with self-generated instructions without the usage of any external annotations by humans or other stronger VLMs.
- **Theoretical and Experimental Analysis**: We present a theoretical analysis from a causal perspective to investigate the mechanisms of self-refinement. Furthermore, we experimentally validate our method using LLaVA-1.5 as the baseline. The dataset, comprising 2

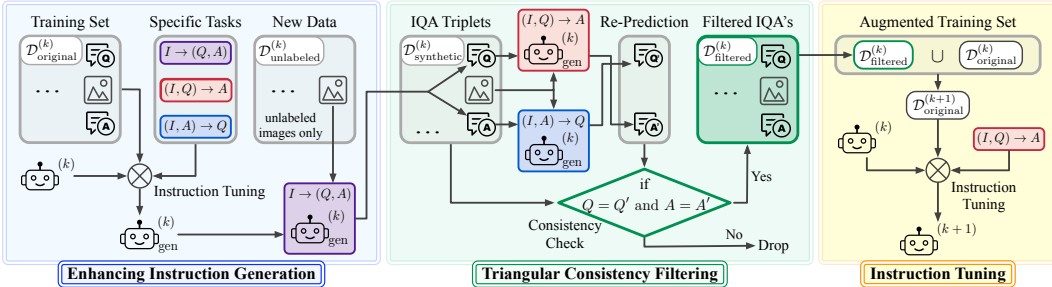

Figure 2: **An overview of the Self-Refinement framework.** The Self-Refinement framework comprises three stages. First, we design multi-task fine-tuning to enhance the model's instruction generation capabilities. Next, we apply the Triangular Consistency principle to filter high-quality instructions based on their scores. Finally, the selected data is used to refine the model. By using the updated model and synthetic data as the starting point for the next iteration, the framework naturally supports an iterative process.

million images accompanied by generated instructions, will be publicly released to facilitate further research.

## 2 Method

In this section, we first define the task and present the overall framework, which consists of three stages. We then detail each stage to demonstrate how the framework facilitates self-refinement in VLMs. Finally, we show that the framework is flexible enough to support multi-round refinement.

### 2.1 Overall framework

Our primary objective is to enable VLMs to refine themselves by extracting supervisory signals from an unlabeled image dataset $\mathcal{D}_{\text{unlabeled}} = \{I_i \mid i = 1, 2, \ldots, N\}$, where N denotes the number of images. To achieve this, we utilize the VLM itself to generate Image-Question-Answer (IQA) triplets for the unlabeled images, thereby constructing a synthetic dataset $\mathcal{D}_{\text{synthetic}} = \{(I_i, Q_i, A_i)\}$. Here, $I_i$ denotes the image, $Q_i$ represents a question generated based on the content of the image, and $A_i$ is the corresponding answer to the question.

As illustrated in Figure 2, we propose a three-stage self-refinement framework. Our preliminary experiments (Table 6) indicate that directly utilizing existing VLMs, such as LLaVA, struggles to generate precise and diverse instructions (question-answer pairs). To address this, we introduce a multitask learning objective in the first stage, incorporating an image-conditional instruction-generation task. Next, a filtering process is applied to collect a refined data set $\mathcal{D}_{\text{filtered}}$, ensuring high-quality supervisory signals. In the final stage, we retrain the VLM using this filtered dataset by tuning the visual instruction.

### 2.2 Enhancing instruction generation

Although current VLMs exhibit strong visual understanding and perform well in answering visual questions, they often lack the ability to generate rich and diverse instructions (e.g., question-answer pairs). To address this limitation, we propose retraining the VLM to generate a synthetic instruction tuning dataset $\mathcal{D}_{\text{synthetic}}$ from an unlabeled image dataset $\mathcal{D}_{\text{unlabeled}}$. For clarity, we denote the original VLM as $M^{(0)}$ and the model after fine-tuning for instruction generation as $M_g^{(0)}$.

Specifically, we modify the typical instruction tuning stage [2] by introducing three specific tasks:

- **I → QA**: Given an image $I_i$, generate a corresponding question-answer pair $(Q_i, A_i)$. Below is an example:

  **Human:** *For this image, what can be the instruction and answer pair?*
  **VLM:** *Instruction: What object is the man holding above his head in the image?*
  *Answer: The man is holding a skateboard above his head in the image.*

- **IQ → A**: Given an image $I_i$ and a question $Q_i$, predict the masked answer $A_i$. This task is the typical instruction tuning task [1]. The following is an example:

  > **Human:** *What other object is the person in the image holding or wearing?*
  > **VLM:** *The person is also holding or wearing a guitar around his neck.*

- **IA → Q**: Given an image $I_i$ and an answer $A_i$, predict the masked question $Q_i$. Here is an example:

  > **Human:** *Build the instruction based on the answer. Answer: Yes, the man in the image appears to be in a band or a performer, as he is holding a guitar around his neck and is in a location that could be a nightclub or performance venue.*
  > **VLM:** *Instruction: Does the man in the image appear to be in a band or a performer?*

Task **I → QA** focuses on enhancing the model's ability to generate instructions effectively. Tasks **IQ → A** and **IA → Q** are aimed at improving data filtering in the subsequent stage by ensuring the consistency and relevance of the generated question-answer pairs.

We preprocess the dataset [2] used in the instruction tuning of the original model $M^0$ to create training data suitable for these tasks. The templates to generate multi-task training data can be found in **Figure A2** of the Supplementary Materials. By organizing the training data in a way similar to an instruction-tuning task, we can directly apply the training strategy used in visual instruction tuning. Formally, the final multi-task loss function for this stage is defined as:

$$\mathcal{L}_{\text{all}} = \mathcal{L}_{\text{qa}} + \mathcal{L}_{\text{a}} + \mathcal{L}_{\text{q}}, \tag{1}$$

where each loss term $\mathcal{L}$ corresponds to one of the tasks and is computed using the cross-entropy loss:

$$\mathcal{L} = -\sum_t \log P\left(w_t \mid V_{\text{instruct}}, w_{<t}\right), \tag{2}$$

where $w_t$ represents the target token at time step $t$. $V_{\text{instruct}}$ is the task-specific instruction including the input information such as the image $I_i$, question $Q_i$, or answer $A_i$, depending on the task.

## 2.3 Triangular consistency filtering

In this stage, we introduce the principle of **Triangular Consistency** to filter high-quality instructions from the generated dataset. This principle is based on a simple hypothesis: robust instructions should demonstrate consistency when any component of an image-question-answer (IQA) triplet is masked and subsequently re-predicted. Specifically, for a given instruction triplet $(I_i, Q_i, A_i)$, where $Q_i, A_i$ are generated by the VLMs:

- If we mask the answer $A_i$ and let the same VLM predict it based on image $I_i$ and question $Q_i$, the predicted answer $A_i'$ should closely match the original $A_i$.
- Conversely, if we mask the question $Q_i$ and predict it using the image $I_i$ and answer $A_i$, the predicted question $Q_i'$ should align with the original $Q_i$.

According to the principle of triangular consistency, the predicted elements $A_i'$ and $Q_i'$ should be consistent with the originals, $A_i$ and $Q_i$, respectively. To quantify this consistency, we define a *consistency score* as:

$$\mathcal{S} = \sqrt{\text{Sim}(Q_i, Q_i') \times \text{Sim}(A_i, A_i')}, \tag{3}$$

where $\text{Sim}(Q_i, Q_i')$ and $\text{Sim}(A_i, A_i')$ represent similarity measures between the original and predicted questions and answers, respectively.

In particular, in order to compute the similarity score across diverse types of question-answer pairs, we have developed appropriate similarity metrics tailored to effectively handle each data type. For single-sentence texts, we utilize a Sentence Transformer [22], while for longer texts, we employ BERTScore [23]. When similarity can be precisely determined, such as in multiple-choice questions, we apply fixed metrics by directly comparing the answers to assess the consistency score. In question-answer pairs involving region descriptions and localization, we measure similarity by calculating the Intersection over Union (IoU).

For each data type, we identify the top 20% of data exhibiting the highest triangular consistency scores. These selected data points form a filtered synthetic instruction tuning dataset, $\mathcal{D}_{\text{filtered}}$, which is then used to refine the VLMs.

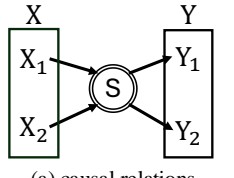 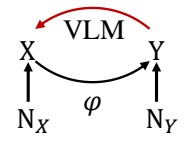

| (a) causal relations | (b) input-output of VLMs |

Figure 3: **Causal modeling and understanding of involved data-generating processes.** Panel (a) presents causal relations among the language $X$, the semantic concept $S$, and the image $Y$. Panel (b) summarizes the input-output relation of VLMs.

## 2.4 Instruction tuning & iteration

After obtaining the synthetic instructions about the unlabeled images, we merge this newly generated data with the original seed dataset (e.g. `llava_v1_5_mix665k` used in LLaVA-1.5 [2]) as the final training dataset. Then we finetune the VLM $M^{(0)}$ on this merged dataset to obtain the self-refined VLM $M^1$. The training strategy is the same as typical instruction tuning as shown in Equation 2.

We demonstrate that this three-stage process can be iteratively applied to enable continuous self-refinement of the model. Starting with the initial VLM $M^{(0)}$ and an unlabeled image set $\mathcal{D}^{(0)}$, we execute the self-refinement procedure as outlined, producing an enhanced model $M^{(1)}$. In the next iteration, we gather additional unlabeled data $\mathcal{D}^{(1)}$ and repeat the process, treating $M^{(1)}$ as the new base model. Through these repeated steps, the model is refined iteratively, resulting in $M^{(2)}$.

By iteratively repeating this process with the vast amount of unlabeled data available online, the model can continually integrate real-world visual information from diverse image distributions. This iterative approach drives progressive self-refinement, allowing the model to adapt to an ever-expanding range of visual data. The self-refinement loop can continue until the VLM has learned from a sufficiently comprehensive data distribution, encompassing most new images within its existing training domain.

## 3 Theoretical analysis

This section explores the self-refinement capability of VLMs from a causal perspective. Specifically, we demonstrate that the language is selected by the semantic concept, and through which, eventually, it serves as a cause of the image. We also present the theoretical foundation for the effectiveness of our self-refinement framework.

### 3.1 Causal relations among language, image, and semantic concept

In order to see the causal relations between the language and the image, let us consider the roles played by the semantic concept $S$ as an auxiliary variable. Such a semantic concept is, in general, regarded as a (latent) common cause of the observations for both image and text modalities [24, 25]. Here, we illustrate the asymmetric relationship between concepts and observations in image $Y$ (with pixels $\{Y_1, Y_2\}$) and text $X$ (with words $\{X_1, X_2\}$). In this context, the semantic concept serves as a selection mechanism for words rather than a common cause, considering the characteristics of natural language. In particular, as illustrated in Figure 3 (a), the semantic concept serves as a *selection* for language, i.e., $\{X_1, X_2\} \rightarrow S$, and as a *(latent) common cause* for image, i.e., $S \rightarrow \{Y_1, Y_2\}$.

Selection mechanisms indicate that samples are chosen based on specific criteria (e.g., only when they meet certain principles) before being observed. Recent literature [26] in language processing explores selection mechanisms, showing that language outputs are modulated by semantic concepts specified in advance. Specifically, the semantic concept that the language needs to convey is a goal to achieve, instead of a common cause for language. The key statistical difference between selection and common cause lies in conditional independence. If a concept acts as a common cause, observations (e.g., words) become more independent when conditioned on that concept. In contrast, selection leads to the opposite effect. Clearly, when conditioned on certain topics, the relations of words would be more dependent, indicating the selection instead of the common cause structure.

Table 1: **Comparison with advanced methods and baselines on 8 vision-language benchmarks.** Due to space limitations, the names of the benchmarks are abbreviated. LLaVA$^W$: LLaVA-Bench (In-the-Wild) [1]; MM-Vet [28]; MME$^C$: MME Cognition; MMB: MMBench [29]; MMB$^{CN}$: MMBench-Chinese; VQA$^{v2}$ [30]; GQA [31]; SQA$^I$: ScienceQA-IMG [32]. More evaluation results on additional benchmarks are provided in **Appendix C.1**. **Data**: Number of Image–Instruction Pairs used during visual instruction tuning. **Recap-LLaVA-1.5** is obtained by using LLaVA-1.5 itself to caption one million images and retraining on those new image–instruction pairs. For comparisons based on the LLaVA-1.5 7B baseline, the best results are **bolded** and the second-best results are underlined.

| Method | LLM | Data | LLaVA$^W$ | MM-Vet | MME$^C$ | MMB | MMB$^{CN}$ | VQA$^{v2}$ | GQA | SQA$^I$ |
|---|---|---|---|---|---|---|---|---|---|---|
| InstructBLIP | Vicuna-7B | 1.2M | 60.9 | 26.2 | - | 36.0 | 23.7 | - | 49.2 | 60.5 |
| IDEFICS-9B | LLaMA-7B | 1M | - | - | - | 48.2 | 25.2 | 50.9 | 38.4 | - |
| Qwen-VL | QWen-7B | 50M | - | - | - | 38.2 | 7.4 | 78.8 | 59.3 | 67.1 |
| Qwen-VL-Chat | QWen-7B | 50M | - | - | 360.7 | 60.6 | 56.7 | 78.2 | 57.5 | 68.2 |
| LLaVA | Vicuna-1.5-7B | 158K | 63.0 | 26.7 | 247.9 | 34.1 | 14.1 | 79.0 | - | 38.5 |
| LLaVA-1.5 13B | Vicuna-1.5-13B | 665K | 70.7 | 35.4 | 295.4 | 67.7 | 63.6 | 80.0 | 63.3 | 71.6 |
| **SRF-LLaVA-1.5 13B** | Vicuna-1.5-13B | 665K + 200k | 73.5 | 37.7 | 334.2 | 68.6 | 64.0 | 81.3 | 65.1 | 72.2 |
| LLaVA-1.5 7B | Vicuna-1.5-7B | 665K | 63.4 | 30.5 | 316.1 | 64.3 | 58.3 | 78.5 | 62.0 | 66.8 |
| Recap-LLaVA-1.5 7B | Vicuna-1.5-7B | 665K + 1M | 63.3 | 29.9 | 321.3 | 66.1 | 58.5 | 79.2 | 62.5 | **69.01** |
| **SRF-LLaVA-1.5** 7B | Vicuna-1.5-7B | 665K + 200k | **66.9** | **33.1** | 331.8 | **66.3** | **59.2** | **79.6** | **63.35** | 67.72 |

## 3.2 Self-Refinement explanation

The input-output relations of a VLM can be summarized as generating the text $X$ based on the visual information $Y$ and conditioned on a question $Q$, represented as $P(X \mid Y; Q)$. To simplify the notation, we fix the question $Q$ and use the shorthand $P(X \mid Y)$ in the following discussions without ambiguity. The initial VLM models the mapping between $Y_{old}$ and $X_{old}$. To further refine the VLM, we incorporate additional unlabeled images, $Y_{new}$, to improve the estimation of $P(X \mid Y)$. We introduce the following mild assumptions, which are widely used in the causality theory [27]:

- **Sufficiency and Independence.** As shown in part (b) of Figure 3, the conditional distribution $P(Y \mid X)$, is defined by a deterministic function $\varphi$ and an independent noise variable $N_Y$, such that $Y = \varphi(X, N_Y)$ with $N_Y \sim P(N_Y)$. Similarly, $X$ is governed by an independent noise variable $N_X$ with $N_X \sim P(N_X)$. The mechanism $\varphi$ is irrelevant to the input distribution $P(X)$.

- **ANM and Decomposition.** The Additive Noise Model (ANM) [33] assumes that the relationship between $X$ and $Y$ can be expressed in the format of $\varphi(X, N_Y) = \varphi(X) + N_Y$. We further posit that the marginal distribution $P(Y)$ can be expressed as the convolution of two distributions:

$$P(Y) = F * G = \int F(z)G(Y - z) \, dz, \tag{4}$$

  where $F$ and $G$ are component distributions. This decomposition is valid under conditions such as $N_Y$ being Gaussian and $P(\varphi(X))$ being indecomposable. The latter assumption is reasonable, as $X$ represents diverse textual data, making $P(\varphi(X))$ unlikely to be decomposable.

Both the independence-of-mechanism and ANM are standard functional assumptions that have been extensively used to identify causal directions and to separate mechanisms from input distributions. In our setting, they provide a principled reason why better estimating the image marginal $P(Y)$—via additional unlabeled images $Y_{new}$—can improve our inference target $P(X \mid Y)$: the forward (causal) model $P(Y \mid X)$ is stable across changes in $P(X)$, whereas the backward direction lacks such invariance. Consequently, unlabeled images carry exploitable signal for refining $P(Y)$ and, through Bayes' rule, for improving $P(X \mid Y)$. We emphasize that Eq. (4) specifies the generative process used for theoretical analysis and is not enforced during VLM training. While these assumptions cannot be directly verified for modern VLMs, they are useful theoretical tools to capture plausible structural properties and to derive interpretable insights.

Our goal is to estimate $P(X \mid Y) = \frac{P(Y|X)P(X)}{P(Y)}$. By incorporating unlabeled images $Y_{new}$, we can refine the empirical estimate of $P(Y)$ with more observations. Since $X$ is observed, improving the estimate of $P(Y \mid X)$ consequently improves our goal $P(X \mid Y)$.

Formally, under ANM assumptions, $P(Y)$ can be expressed as a convolution of the distributions of $\varphi(X)$ and $N_Y$. From the paired data $(X_{old}, Y_{old})$, we can estimate the function $\varphi_{old}$ and corresponding

distribution $P(\varphi_{old}(X))$. Having estimates of both $P(Y)$ and $P(\varphi_{old}(X))$, we can identify the noise distribution $P(N_Y)$ via deconvolution: $P(N_Y) = P(Y) * P(\varphi_{old}(X))^{-1}$. Due to the unique decomposition of $P(Y)$, we can identify whether $P(N_Y)$ corresponds to $F$ or $G$, and thus get the true distribution $P(N_Y)$. This correction of $P(N_Y)$ can help obtain a better estimation of $P(Y \mid X)$.

# 4 Experiments

## 4.1 Experimental settings

We adopted LLaVA-1.5 7B [2] as our baseline, which uses CLIP-Large [34] as visual encoder and Vicuna-1.5 [35] as LLM. To rule out the possibility that our gains stem merely from more data or longer training, we constructed an additional strong baseline, Recap-LLaVA-1.5, which uses the same unlabeled images but replaces the Self-Refinement procedure with a straightforward self-annotation pipeline: (i) starting from the same 1 M unlabeled images as SRF-LLaVA-1.5, we use a frozen LLaVA-1.5-7B to generate one detailed caption per image; (ii) we retain all 1 M captions without filtering and merge them with the original LLaVA-1.5 training set; and (iii) we fine-tune the model using the identical training schedule as SRF-LLaVA-1.5.

Beyond the original training dataset `llava_v1_5_mix665k` used in LLaVA-1.5, we randomly selected 2.8 million images from LAION [36] as the unlabeled image set to validate the self-refinement ability. During fine-tuning on the merged dataset, we follow the standard LLaVA-1.5 instruction-tuning recipe. Specifically, we keep the vision encoder frozen and fine-tune both the image-text projection layer and the LLM parameters. In the evaluation, we followed the setup of LLaVA-1.5, which contains 8 benchmarks. For traditional VQA tasks, we used the VQAv2 [30], GQA [31], and ScienceQA [32] datasets. For visual perception and reasoning tasks, we employed MMBench [29], MMBench-Chinese [29], MME [37], and MM-Vet [28] benchmarks. To assess visual dialogue ability, we utilized the LLaVA-Bench(In-the-Wild) [1] benchmark. Please note that the goal of this work is to explore the self-refinement capability of VLMs without distilling knowledge from other models like GPT-4V. Therefore, we did not incorporate such knowledge into our model and did not conduct many comparisons with the methods using such models.

## 4.2 Implementation details

In the Enhancing instruction generation stage, we constructed the dataset using `llava_v1_5_mix665k` as the seed. Specifically, we masked both the question (Q) and the answer (A) in 50% of the data, only Q in 20%, and only A in the remaining 30%. For each task, we randomly selected prompt templates combined with unmasked elements as inputs, with the masked parts serving as ground-truth targets. Then we generate synthetic data from one million unlabeled images and select the top 20% ones with higher Triangular Consistency scores. We design a hybrid scoring approach suited to different instruction types. For shorter texts, we employed UAE-Large-V1[22], a compact Sentence Transformer with only 335 million parameters, that effectively compares single-sentence text similarity without introducing external knowledge. For longer texts, since the performance of the Sentence Transformer diminishes, we opt for BertScore[23], based on the bert-base-uncased model, as it provides robust discriminative capabilities for lengthier text comparisons. To ensure fair comparisons with baseline, we adopted a head-to-head setup by using the same base language model and prompts as LLaVA-1.5. Also, we adhered strictly to the prompt templates and data preprocessing methods utilized by LLaVA-1.5. The specific calculation procedure for Triangular Consistency scores is provided in **Section B** of Supplementary Materials.

## 4.3 Quantitative evaluation

We evaluated our framework alongside advanced VLMs, the baseline LLaVA-1.5 and the strong baseline SRF-LLaVA-1.5 on 8 benchmarks. Overall, our framework offers consistent improvements over LLaVA-1.5 on most of the benchmarks. For instance, in traditional VQA tasks—including VQAv2 and GQA, our model outperformed LLaVA-1.5 by an average of 1.22%. For visual perception and reasoning tasks such as MMBench, MM-VET, and LLaVA-Wild, we achieved improvements of 2.1%, 1.7%, and 3.5%, respectively. Although Recap-LLaVA-1.5 shows only small gains on most benchmarks, SRF-LLaVA-1.5 improves them much more. This suggests that SRF-LLaVA-1.5's

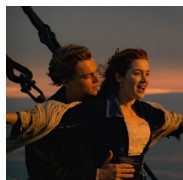

**Question**: What's the ending of this movie?

**LLaVA-1.5**:The ending of this movie is not provided, as it is a scene from a movie or a TV show. However, it is likely that the couple will continue their adventure, as they are already on a boat and enjoying their time together.

**Ours:**The ending of the movie Titanic is well-known, as it is a classic romantic drama. In the film, the main characters, Jack and Rose, are on a ship called the Titanic. They fall in love and share a memorable night together on the deck of the ship. However, as the ship encounters an iceberg and begins to sink, Jack sacrifices his life by giving Rose a chance to survive by jumping into the icy water. The movie ends with Rose being rescued and the sinking of the Titanic. The emotional impact of the film and the tragic ending have made it a beloved classic and a symbol of love and sacrifice.

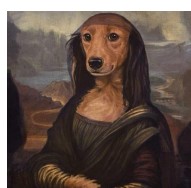

**Question**: Describe this photo in detail.

**LLaVA-1.5**: The image features a painting of a dog wearing a Renaissance-style dress, giving it a human-like appearance. The dog is sitting in a chair, looking directly at the viewer. The painting is quite detailed, capturing the dog's facial expression and the intricate design of the dress. In the background, there is a person partially visible on the left side of the painting. The overall scene is quite captivating and unique, as it combines the artistic style of the Renaissance with the subject matter of a dog.

**Ours:** The image features a painting of a dog dressed in a Renaissance-style outfit, resembling a woman. The dog is wearing a long dress and a hat, giving it a unique and artistic appearance. The painting is set in a landscape with a mountainous background, adding to the overall ambiance of the scene. The dog is positioned in the center of the painting, with its hands resting on its lap, as if it were a human. The painting captures the dog's unique and creative portrayal, making it an interesting and eye-catching piece of art.

Figure 4: **Two comparison examples between our SRF-LLaVA-1.5 and LLaVA-1.5 in visual chat**. Red highlights indicate factual errors or irrelevant content in the response, while green highlights emphasize image details critical for providing an accurate answer.

Table 3: **Evaluations on MoblieVLM Baseline**

| Method | LLM | GQA | SQA$^I$ | VQA$^T$ | POPE | MME$^P$ | MMB |
|---|---|---|---|---|---|---|---|
| **MobileVLM** | MobileLLaMA-1.4B | 56.1 | 57.3 | 41.5 | 84.5 | 1196.2 | 53.2 |
| **SRF-MobileVLM** | MobileLLaMA-1.4B | **58.1** | **59.8** | **43.3** | **85.3** | **1220.3** | **56.1** |

advantage comes mainly from the Triangular Consistency principle, which enhances self-refinement, rather than from training on extra data.

**Training cost analysis.** At inference time, SRF-LLaVA-1.5 and LLaVA-1.5 are identical in parameter count and therefore require the same FLOPs and GPU memory. During training, however, SRF-LLaVA-1.5 processes a larger number of images, leading to a longer training duration. Table 2 compares the total wall-clock times of SRF-LLaVA-1.5 and LLaVA-1.5, both trained on a cluster equipped with 8 NVIDIA H100-NVL GPUs (96 GB each). The refinement procedure costs approximately 40 additional GPU-hours ($\approx 0.7\times$), which we report explicitly to inform practitioners of the trade-off between accuracy and computational expense.

Table 2: **Training time comparison between LLaVA-1.5 and SRF-LLaVA-1.5.**

| Model | Total wall-clock time | GPU-hours |
|---|---|---|
| LLaVA-1.5 | 7 h | 56 |
| SRF-LLaVA-1.5 | 12 h | 96 |

## 4.4 Qualitative evaluation

In this section, we illustrate how our self-refinement framework enhances the real-world visual dialogue capabilities. Figure 4 presents two representative examples.

**Enhanced World Knowledge.** For the first example in Figure 4, our model correctly identifies the individuals in the image as the protagonists of the movie *Titanic* and accurately describes the film's ending, whereas the original LLaVA-1.5 fails to do so. Though no external information is involved, our framework can use the consistency principle to filter possible errors and do the refinement.

**Improved Generalization Ability.** For the second example, our model generates a description that is significantly more detailed and accurate than that of LLaVA-1.5, without exhibiting hallucinations. We attribute this improvement to the exposure to a wider variety of visual scenes and objects provided by the unlabeled images. By learning from these diverse visual features during training, the model enhanced its ability to recognize and represent them.

## 4.5 Ablation studies

**Does Self-Refinement Generalize Across Different Scales and Architectures of VLMs? Yes.** To assess the generalization of our Self-Refinement framework across VLMs with different parameter scales and architectures, we conducted experiments on MobileVLM-1.7B [38], QWen2.5-VL 3B [39],

Table 4: **Evaluations on QWen2.5-VL Baseline**

| Method | LLM | MMMU | MMMU Pro | MathVista | MathVision | MMStar | MMB |
|---|---|---|---|---|---|---|---|
| **Qwen2.5-VL** | Qwen2.5-3B | 53.1 | 31.6 | 62.3 | 21.2 | 55.8 | **81.5** |
| **SRF-Qwen2.5-VL** | Qwen2.5-3B | **55.0** | **31.9** | **64.7** | **23.3** | **56.5** | 80.8 |

and LLaVA-1.5 13B using an identical dataset and training pipeline. For MobileVLM and QWen2.5-VL 3B, we followed their original configurations and, for each model respectively, evaluated SRF-MobileVLM and SRF-QWen2.5-VL on six benchmarks selected from their official evaluation suites. For LLaVA-1.5 13B, we mirrored the experimental setup used for LLaVA-1.5 7B. As shown in Table 3 and Table 4, SRF delivers consistent improvements in all baselines, indicating that self-refinement is generalized across various VLM architectures. In particular, as shown in Table 1, SRF-LLaVA-1.5 13B exceeds LLaVA-1.5 13B in all 8 benchmarks, demonstrating effectiveness at larger parameter scales.

**Can Multiple Iterations Lead to Better Performance? Yes.** As shown in Table 5, Round 2 performance was comparable to Round 1 on GQA and SQA, with noticeable improvements on MM-Vet and LLaVA-Bench. However, the overall improvement in Round 2 was smaller. We hypothesize this saturation stems from the finite exploitable signal that unpaired images provide about the underlying image distribution; once largely distilled, subsequent iterations yield diminishing returns. Nevertheless, a second round still produced measurable gains, implying that a single pass does not fully harvest the available signal. Round 2 exhausted all our collected images, suggesting that further performance gains would require additional data.

Table 5: **Ablation study on Framework Iteration, Consistency Criterion, Selection Threshold.** Complete results are provided in **Table A3** of Supplementary Materials.

| Ablations | Patterns | GQA | SQA$^I$ | MM-Vet | LLaVA$^W$ |
|---|---|---|---|---|---|
| **SRF-LLaVA-1.5** | Default | 63.35 | 67.72 | 33.1 | 66.9 |
| **LLaVA-1.5** | Baseline | 62.00 | 66.8 | 30.5 | 63.4 |
| **Multi-Round** | Round-2 | 63.25 | 67.77 | **33.9** | **67.7** |
| **Consistency** | Bottom 20% | 62.85 | 66.70 | 30.4 | 64.4 |
| **Threshold** | Top 5% | 62.90 | 67.48 | 32.4 | 63.6 |
| | Top 20% | **63.35** | 67.72 | 33.1 | 66.9 |
| | Top 50% | 63.02 | **68.77** | 30.4 | 62.8 |
| | Top 80% | 62.9 | 67.03 | 29.1 | 64.0 |
| | Top 100% | 62.87 | 67.48 | 30.7 | 63.7 |

**Does the Self-Refinement Framework Produce Better Instructions? Yes.** Table 6 compares QA pairs generated by our Self-Refinement framework against those from prompted original LLaVA-1.5, evaluating GPT-4o accuracy and diversity metrics (TTR [40] and Distinct-2 [41]) on 1000 samples. Our model demonstrates clear advantages over prompted LLaVA-1.5, achieving superior accuracy and diversity. This indicates that explicitly training a dedicated model for instruction generation effectively improves QA pair quality, particularly in diversity.

Table 6: **Evaluations of GPT-4o accuracy, Type-Token Ratio (TTR), and Distinct-2 on the generated QA pairs from SRF-LLaVA, baseline LLaVA, and LLaVA with in-context learning.**

| | SRF-LLavA 7B | LLaVA-1.5 7B | | |
|---|---|---|---|---|
| **Metrics** | 0-shot | 0-shot | 1-shot | 5-shot |
| **Acc(%)** | **85.3** | 73.9 | 79.2 | 63.6 |
| **TTR** | **0.1144** | 0.0888 | 0.0775 | 0.0692 |
| **Distinct-2** | **0.4790** | 0.3216 | 0.3075 | 0.2737 |

**How does Triangular Consistency Principle work?** To evaluate the effectiveness of the Triangular Consistency Filtering stage, we conducted two comparisons: (1) between the retained and excluded QA pairs after filtering and (2) by altering the filter criterion from the top 20% to the bottom 20%. Table 7 presents both quantitative and qualitative evaluations of the retained versus excluded synthetic data subsets. For the quantitative analysis, we used GPT-4o to evaluate 1,000 samples from each subset. We also conducted a human study with 12 volunteers, each of the 100 samples reviewed by 3 volunteers. The results confirm that the retained subset significantly outperforms the excluded subset. Qualitative comparisons of the QA pair samples at the bottom of Table 7 further emphasize the superior quality of data selected by the Triangular Consistency Principle. Additionally, as shown in the "Consistency" group of Table 5, training the model on the lower 20% (ranked with consistency score) of synthetic data resulted in a performance decline compared to SRF-LLaVA-1.5 (which using top 20%), further validating the effectiveness of Triangular Consistency. More experimental details on Triangular Consistency Filtering can be found in **Section C** of Supplementary Materials.

**How do Different Instruction Types Influence Performance?**

To further investigate which data types have the greatest impact on model performance, we replaced the filtered synthetic data with different subsets of data categories. Specifically, we retained only five types including VQA (Visual Question Answering), Visual Chat, REG&REC (Region Expression and Recognition), Caption, and Multiple-Choice/Judgment, and retrained LLaVA-1.5 exclusively on these subsets to observe any changes in downstream performance. As shown in Figure 5, different instruction types yield corresponding improvements, with captioning tasks contributing the most overall enhancement. Please refer to **Section C** of Supplementary Materials for comprehensive comparisons, statistical details on the distribution of generated instructions, and further experiments on the impact of different instruction partitions.

Table 7: **Quantitative and Qualitative Analysis of Retained and Excluded QA Pairs.** We evaluate the quality of the generated QA pairs through both GPT-4o assessments and a human user study. Additionally, we present an example to qualitatively compare the consistency between the Original and Recovered QA pairs.

| Type | Evaluation / Examples | | |
|---|---|---|---|
| **Quant.** | **Subset** | **GPT** | **Human** |
| | Retained | **85.3** | **89.0** |
| | Excluded | 59.9 | 83.0 |
| | **Subset** | **Original QA** | **Recovered QA** |
| **Qual.** | Retained | **Q**: What is the color of the painting displayed in the image? **A**: The painting displayed in the image is a blue picture. | **Q'**: What color is the picture displayed in the image? **A'**: The color of the painting displayed in the image is blue. |
| | Excluded | **Q**: What is the setting of the image? **A**: The setting of the image is indoors, with the person using a green case in a white room. | **Q'**: What is the setting of the image? **A'**: The setting of the image is outdoors, with the person holding the small case or container outside. |

**How Much Data is Required?** More data offers additional information but also introduces more noise. We investigated the impact of varying the threshold of the triangular consistency score used to screen high-quality synthetic data. Specifically, we combined synthetic data with scores ranking in the top 5%, 20%, 50%, 80%, and 100% (i.e., without filtering and detailed captions) with the LLaVA-1.5 665k dataset to form the new training set, as illustrated in Table 5. The results indicate that model performance improves across all thresholds. However, the best overall performance is achieved when using the top 20% threshold.

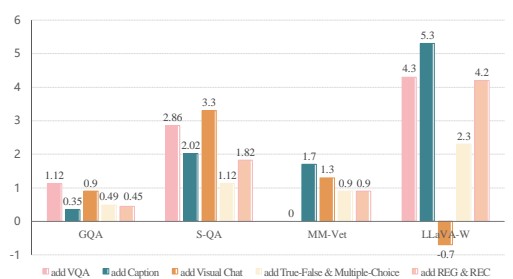

Figure 5: **Ablation study on specific synthetic data types.** The numbers denote the percentage increase compared to LLaVA-1.5.

## 5 Conclusion

In this work, we explored the self-refinement capability of Vision-Language Models (VLMs) through a framework grounded in the Triangular Consistency principle. Our results demonstrate that this self-refinement ability can be effectively activated, enabling VLMs to autonomously generate and leverage high-quality supervision from unlabeled data. We further provided a causal explanation for this phenomenon, showing how the model's ability to infer missing modalities supports its internal consistency and learning dynamics. Empirical evaluations confirm that our framework consistently improves baseline performance without external supervision, highlighting the potential of self-refinement as a pathway toward autonomous and continually improving multimodal intelligence. **Limitations:** Although our approach successfully demonstrates self-refinement, the model's improvement diminishes when the newly introduced images closely align with the distribution of the model's existing training data. **Societal Impact:** By enabling self-improvement from unlabeled images, our framework can reduce reliance on expensive and labor-intensive human annotations and help democratize access to high-performance VLMs, particularly in low-resource settings. However, learning from model-generated synthetic data risks reinforcing or amplifying biases and hallucinations present in the pretrained model, and the absence of human-in-the-loop oversight can make unintended behaviors harder to trace or audit.

## Acknowledgment

We would like to acknowledge the support from NSF Award No. 2229881, AI Institute for Societal Decision Making (AI-SDM), the National Institutes of Health (NIH) under Contract R01HL159805, and grants from Quris AI, Florin Court Capital, and MBZUAI-WIS Joint Program, and the Al Deira Causal Education project.

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

# Appendix of "Towards Self-Refinement of Vision-Language Models with Triangular Consistency"

**Yunlong Deng**[1*], **Guangyi Chen**[1,2*], **Tianpei Gu**[3], **Lingjing Kong**[2], **Yan Li**[1], **Zeyu Tang**[2], and **Kun Zhang**[1,2]

[1]Mohamed bin Zayed University of Artificial Intelligence
[2]Carnegie Mellon University
[3]ByteDance US

Appendix organization:

# A  Related Work

**Vision Language Models.** Vision-Language Models (VLMs) leverage large datasets to learn joint representations of visual and linguistic information. These models commonly employ training frameworks such as contrastive matching [42, 43, 44, 45, 46, 47, 48, 49], cross-modal generation [50, 51, 52, 53, 54, 55, 56, 57], or a combination of both. Key examples include contrastive frameworks like CLIP [58] and ALIGN [59] which synchronize text and image representations through contrastive objectives. Conversely, the cross-modal generation approach, exemplified by works such as VirTex [50] and VisualBERT [52], emphasizes generative tasks like captioning or masked content reconstruction for learning representations. Additionally, the emergence of large language models has significantly enhanced the capabilities of multimodal systems. These methods [12, 1, 2, 4, 6, 13, 60] integrate and expand upon this rich multimodal knowledge by aligning the pre-trained LLMs with the visual modality.

**Visual Instruction Tuning with Synthetic Data.** To address the high costs and complexities of acquiring copyrighted visual instruction data, recent approaches [8, 9, 10, 11, 61, 62] have advocated for the use of synthetic data in model training. These methods typically utilize advanced VLMs such as GPT4V [12] to analyze requirements, generate detailed captions or instructions, and assess synthetic data quality. For instance, ShareGPT4V [8] and LLaVA-Next [9] employ GPT4V to produce captions and instructions, respectively, while MLLM-DataEngine [11] focuses on identifying model weaknesses and generating instructions with specific targeting. While these methods enhance performance, they encounter two major limitations tied to the reliance on advanced VLMs: firstly, high-quality VLMs often come with restrictive usage limits and high costs; secondly, as models evolve, sourcing superior teacher models for effective data annotation becomes increasingly difficult.

**Self-Refinement of LLMs.** While discussions on VLMs' self-refinement are limited, debates about LLMs' self-refinement persist. These methods fall into two categories: extrinsic and intrinsic self-refinement. Extrinsic methods utilize feedback from external sources, such as evaluation models [20, 19, 63, 21, 64] or interactions with humans or tools [65, 66, 67, 68]. More related to this work is the intrinsic self-refinement [69, 18, 70, 71, 72], which utilizes the model's capabilities without external feedback. For instance, Self-Refine [69] employs the in-context learning of LLMs to generate and iteratively correct instructions using seed examples. Sun et al. [15] apply templates and text-based rules for instruction selection, while IoE Prompt [72] uses carefully designed prompts for self-correction through re-inference. However, applying these methods to VLMs presents significant challenges. Multiple inferences [72] and multi-example in-context learning [69] are resource-intensive with images. In addition, the use of image modality means that sample templates or text-only rules, which only focus on textual quality without considering alignment, are less effective.

# B  More Implementation Details

In this section, we present some key implementation details of the Self-Refinement Framework to facilitate a better understanding of its mechanisms.

## B.1  Templates for Generating Multi-Task Training Data

We provide the prompts used to generate multi-task training data. During the training phase of Enhancing Instruction Generation, we transform the original training data from the LLaVA-1.5 database, `llava_v1_5_mix_665k`, into data suitable for multi-task training by setting specific prompt templates. Specifically, for the Image-Question-Answer (IQA) data triples in original dataset, we established the following task types: I → QA, IQ → A, and IA → Q.

For the IQ → A task, we use the original data without modification. For the I → QA and IA → Q tasks, we prepend a randomly selected prompt to the inputs to assist the VLM in more accurately aligning the instructions with the respective downstream tasks. Additionally, we have also redesigned the system prompts used by the VLM due to changes in training tasks. The complete prompts used for training are detailed in Figure A1.

- You are a visual-language assistant. You will be given one of the three kinds of tasks with specified task type. Complete the task by generating a response.

- Generate the instruction and answer pair corresponding to the image.
- Build the instruction and answer pair for this image.
- What can be the question and answer pair corresponding to the image?
- For this image, what can be the instruction and answer pair?
- Given the image, generate the question and answer pair.
- Can you generate the question and answer pair for this image?

- Generate the instruction based on the answer.
- Build the instruction based on the answer.
- What can be the instruction for the answer?
- What's the question for this answer?
- For the response, what can be the instruction?

Figure A1: **The list of prompts for Enhancing Instruction Generation.** Given that IQ → A task is the same as typical instruction tuning, We provide the system prompt and template prompts for I → QA, and IA → Q tasks. Notably, we do not enforce any particular question–answer type. The prompts simply ask the model to generate a question and its answer. The distribution of generated QA types is therefore emergent

## B.2 Similarity Calculation across Diverse Data Types

We demonstrate detailed information on the similarity indices designed for different types of data during the Triangular Consistency Filtering stage. We process various types of question-answer data generated during the previous stage and flexibly adjust the similarity calculations based on data characteristics, as follows:

- To assess text similarity, we use either Sentence Embedding or BertScore. As detailed in Section 2.3, we separately calculate the similarities of questions $(Q, Q')$ and answers $(A, A')$, then compute their geometric mean. For Visual Question Answering (VQA) data, we employ Sentence Embedding for both questions and answers. In contrast, for Visual Chat data with longer answers, we apply Sentence Embedding to questions and use BertScore for answers, as it performs better with extended texts.

- When similarity values can be precisely calculated, we compute them directly using specific metrics. For example, in Region Description (REG) and Recognition (REC) data involving a region description query (Q) and corresponding coordinates (A), or vice versa, we use Intersection over Union (IoU) to measure the similarity of region coordinates. The similarity of region descriptions is calculated using Sentence Embedding, and we take the geometric mean of both similarities. Similarly, for multiple-choice and true-false data consisting of only a question and an answer, we do not compare question similarity since the same answer might correspond to different questions. Instead, we check if the answer remains the same before and after reconstruction; if identical, the similarity score is 1; otherwise, it is 0.

**Visual Chat**
**Question:** Why might each of the three shots in this photograph be taken?
**Answer:** Each of the three shots in the photograph captures a different high angle angle and side view of a man wearing a suit and tie, possibly for a wedding, as one of the angles suggests. These shots might be taken to showcase the suit, tie, and overall look from different angles and perspectives, providing a more comprehensive and visually appealing coverage of the desired outfit. The variations in camera angles also provide a sense of uniqueness and individuality, helping the viewer better understand the significance of the clothing and the overall attire being presented.

**REG** & **REC**
**Question:** Please provide the bounding box coordinate of the region this sentence describes: the mans right arm.
**Answer:** [0.82, 0.83, 0.97, 0.98]
**Question:** Please provide a short description for this region: [0.67, 0.78, 0.8, 0.93].
**Answer:** Two buttoned up shirts.

**Caption**
**Question:** Provide a one-sentence caption for the provided image. Reference OCR token: Book, BEST
**Answer:** A poster of a smiling cartoon character in a white and blue shirt with the words best at the bottom of the photo.

**Multiple-Choice** & **True-False**
**Question:** What kind of bag is shown? A. backpack B. tote C. purse D. briefcase Answer with the option's letter from the given choices directly.
**Answer:** B
**Question:** Is there a balcony in the picture? Answer the question using a single word or phrase.
**Answer:** Yes

Figure A2: **The list of Q-A pairs of different types.** The parts marked in orange are fixed prompt templates, which are not included in similarity calculations.

- For data where the questions are in a fixed format, we only compute the similarity of the answers. For example, for Caption data, which includes an instruction (Q) and its corresponding image description (A), we focus solely on the similarity of the image descriptions. This is because the instructions primarily contain hints and OCR tokens without significant meaning.

To fairly treat different types of data, we select the top 20% of data from each category. Additionally, to prevent the similarity calculations from being influenced by repeated instruction templates, we exclude all fixed-format instructions. Examples of each data type are shown in Figure A2.

### B.3 Hyperparameters

To ensure a fair comparison, we adopted a head-to-head setup by using the same base language model and prompts as LLaVA-1.5. The training hyperparameters were consistent with those of LLaVA-1.5: an initial learning rate of $2 \times 10^{-5}$ and a batch size of 128. Most of the training (including all ablation studies) was conducted on 8 NVIDIA A100 GPUs, with both the first and third stages

each taking approximately 20 hours. For evaluation, we compared our **SRF-LLaVA-1.5** against the original LLaVA-1.5. We adhered strictly to the prompt templates and data preprocessing methods utilized by LLaVA-1.5. Decoding was performed using greedy search to maintain consistency and reproducibility in our results.

### B.4 Training Details of Other Baseline VLMs

The merged dataset for MobileVLM consists of (i) samples generated by MobileVLM itself through our self-refinement loop on the 1M unlabeled images, and (ii) its original supervised training set. For QWen2.5-VL, since its original training set is not publicly available, we follow the official repository's recommendations by combining our self-refinement dataset with the officially suggested datasets [73] (nyu-visionx/Cambrian-10M, lmms-lab/LLaVA-NeXT-Data, FreedomIntelligence/ALLaVA-4V, and TIGER-Lab/VisualWebInstruct) and subsequently perform SFT on the combined corpus.

### B.5 Pseudo Code

---
**Algorithm A1** Iterative Self-Refinement
---
1: **Input:** $M_0$ ... initial VLM
2:              $D_0$ ... human-labeled $(I, Q, A)$ dataset
3:              $U$ ... unlabeled images
4:              $K$ ... number of refinement rounds
5: **for** $k = 1$ **to** $K$ **do**
   $\triangleright$ —— *iterative self-refinement* ——
   **Stage 1 – Multi-task fine-tune (caption, VQA, instruction)**
6:       $M_0^{\text{gen}} \leftarrow \text{TRAIN}(M_0, D_0)$
7:       $S \leftarrow \{ (I, Q, A) \text{ produced by } M_0^{\text{gen}} \text{ for every } I \in U \}$
8:       $S' \leftarrow \{ (I, Q', A') \mid Q' \leftarrow M_0^{\text{gen}}(I, A) \wedge A' \leftarrow M_0^{\text{gen}}(I, Q) \}$
   **Stage 2 – Generate & filter synthetic IQA**
9:       **for all** $(I, Q, A) \in S$ **and** $(I, Q', A') \in S'$ **do**
10:          **if** $Q = Q'$ **and** $A = A'$ **then**
11:              $F \leftarrow (I, Q, A)$
12:          **end if**
13:      **end for**
   **Stage 3 – Instruction tuning with filtered data**
14:      $D_1 \leftarrow D_0 \cup F$
15:      $M_1 \leftarrow \text{TRAIN}(M_0, D_1)$                    $\triangleright$ instruction-only objective
16: **end for**
17: **return** $M$
---

As shown in Algorithm A1, we provide the pseudo code for the core algorithm of our SRF-LLaVA-1.5. This pseudo code serves as a high-level description of the algorithm's workflow, outlining its core computational steps and logical structure.

## C  More Quantitative Results

### C.1  Evaluation on More Challenging Benchmarks

To evaluate whether our method generalizes to more challenging benchmarks, we further assess SRF-LLaVA-1.5 7B and LLaVA-1.5 7B on MMMU and MMMU-Pro. The results are shown in Table A1.

### C.2  Ablation Results on More Benchmarks

Table A1: **Comparison on more challenging benchmarks.** For both LLaVA-1.5-7B and SRF-LLaVA-1.5-7B, we obtained the results through local evaluation using VLMEvalKit.

| Model | MMMU | MMMU Pro |
|---|---|---|
| Baseline | 34.7 | 17.6 |
| After refinement | **36.6** | **18.3** |

This section presents the complete ablation study data conducted within the Self-Refinement Framework. We examined the impact of various factors on the framework's performance, including the number of framework iterations, the selection of consistency criteria, the setting of filtering thresholds, and the adjustment of data distribution. As shown in Table A2, we observed the following phenomena:

Table A2: **Complete ablation study results on Framework Iteration, Consistency Criterion, Selection Threshold, and Data Partition.** LLaVA$^W$: LLaVA-Bench (In-the-Wild) [1]; MM-Vet [28]; MME$^C$: MME Cognition [37]; MMB: MMBench [29]; MMB$^{CN}$: MMBench-Chinese; VQA$^{v2}$ [30]; GQA [31]; SQA$^I$: ScienceQA-IMG [32].

| Ablations | Patterns | LLaVA$^W$ | MM-Vet | MME$^C$ | MMB | MMB$^{CN}$ | VQA$^{v2}$ | GQA | SQA$^I$ |
|---|---|---|---|---|---|---|---|---|---|
| **SRF-LLaVA-1.5** | Default | 66.9 | 33.1 | 331.8 | 66.3 | 59.2 | 79.6 | 63.35 | 67.72 |
| **LLaVA-1.5** | Baseline | 63.4 | 30.5 | 316.1 | 64.3 | 58.3 | 78.5 | 62.0 | 66.8 |
| **Multi-Round** | Round2 | 67.7 | 33.9 | 335.0 | 66.3 | 60.3 | 79.52 | 63.25 | 67.77 |
| **Consistency** | Lower Score | 64.4 | 30.4 | 337.5 | 66.0 | 60.3 | 79.19 | 62.85 | 66.7 |
| **Partition** | without long caption | 64.8 | 30.1 | 321.4 | 66.3 | 60.2 | 79.41 | 63.27 | 67.03 |
| | LLaVA-1.5 partition | 62.2 | 30.2 | 282.1 | 66.8 | 60.0 | 79.42 | 63.17 | 68.12 |
| **Threshold** | top 5% | 63.6 | 32.4 | 337.5 | 65.8 | 61.2 | 79.34 | 62.9 | 67.48 |
| | top 20% | 66.9 | 33.1 | 331.8 | 66.3 | 59.2 | 79.6 | 63.35 | 67.72 |
| | top 50% | 62.8 | 30.4 | 319.3 | 67.1 | 60.9 | 79.44 | 63.02 | 68.77 |
| | top 80% | 64.0 | 29.1 | 316.4 | 66.5 | 60.4 | 79.33 | 62.9 | 67.03 |
| | top 100% | 63.7 | 30.7 | 312.5 | 65.4 | 59.4 | 79.35 | 62.87 | 67.48 |

Table A3: **Consistency score between reference and candidate answers.**

| Type | Answer | Similarity |
|---|---|---|
| **Ref** | The answer is an apple. | – |
| **Cand1** | The answer is an orange. | 0.79 |
| **Cand2** | The answer happens to be an apple. | 0.98 |
| **Cand3** | An apple is the correct answer. | 0.94 |

**Improved Performance with Additional Iterations**    With more iterations, the model's performance improves. The Round-2 model slightly surpasses SRF-LLaVA-1.5 on 5 benchmarks and is comparable on others, achieving an average improvement of 0.4% across all benchmarks (excluding MME-C due to unnormalized values, and keeping this setting for all following discussions). The performance gain is more significant on real-world visual chat benchmarks than on traditional VQA tasks, suggesting that supervisory information from unlabeled images in the new distribution benefits real-world dialogue scenarios.

**Preserving Data Distribution and Detailed Captions Improves Performance**    Maintaining the original distribution of synthetic data, along with detailed captions, enhances self-refinement performance. Adjusting the data distribution to match that of the LLaVA-1.5 training set causes a slight performance drop. Likewise, removing all long-caption data decreases performance.

**Impact of Filtering Threshold**    The filtering threshold affects model performance. A 20% threshold yields the best results among thresholds of 5%, 20%, 50%, and 80%, suggesting the importance of balancing data quality and quantity. Even without filtering (threshold of 100%), the model shows some performance improvement (about 0.73%). This indicates that the VLM can learn supervisory information from new images, even with low-quality synthetic annotations.

### C.3   Effectiveness of Consistency Score Measurement

We used to explore several consistency score measurements, including RoBERTa and Vicuna1.5, which yielded lower accuracy and higher costs, so we adopted the current method. Specifically, we employ Sentence Transformer for shorter sentences, while for longer sentences or paragraphs, we utilize BertScore, ensuring an optimal balance between evaluation accuracy and computational efficiency. Table A3 presents consistency scores for sentence meaning shifts due to word changes. The score is notably lower when the factual word "apple" changes to "orange."

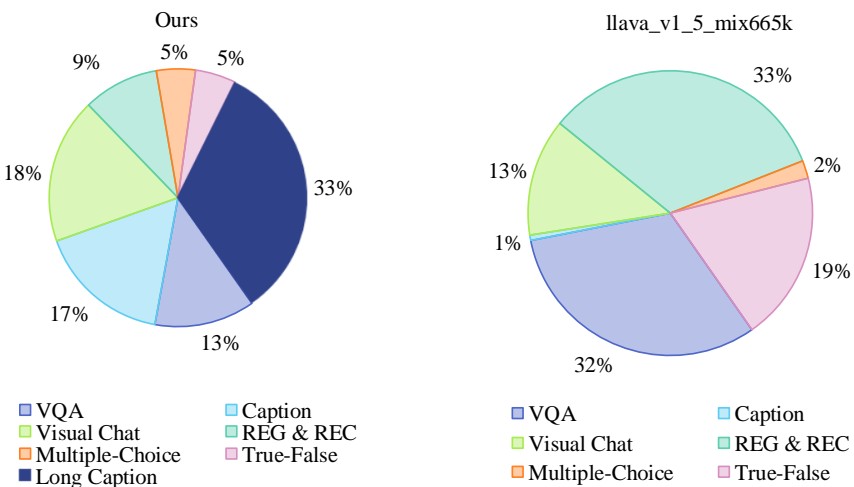

Figure A3: **The comparison between the data distributions of synthetic data and LLaVA-1.5 training dataset.**

Table A4: **Complete ablation study results on specific synthetic data types**. Each value in the *Data Partition* column corresponds to retaining *only* the indicated synthetic–data type.

| Data Partition | LLaVA$^W$ | MM-Vet | MME$^C$ | MMB | MMB$^{CN}$ | VQA$^{v2}$ | GQA | SQA$^I$ |
|---|---|---|---|---|---|---|---|---|
| Add VQA | 68.7 | 31.1 | 322.5 | 65.8 | 59.5 | 79.28 | 63.12 | 69.66 |
| Add Caption | 63.5 | 33.0 | 326.4 | 67.2 | 61.4 | 79.20 | 62.90 | 70.10 |
| Add Visual Chat | 69.5 | 32.8 | 335.4 | 66.4 | 60.5 | 79.27 | 62.35 | 68.82 |
| Add REG&REC | 68.4 | 33.0 | 287.9 | 66.6 | 59.5 | 79.20 | 62.45 | 68.62 |
| Add True-False/Multiple-Choice | 66.5 | 32.0 | 309.3 | 66.8 | 59.0 | 79.09 | 62.49 | 67.92 |

## C.4 Data Types Investigation

This section presents the complete results of ablation studies on specific synthetic data types. This experiment aims to comprehensively assess the impact of integrating different data types on the Self-Refinement performance of VLMs.

Figure A3 illustrates the proportional differences in distribution between our synthetic data and the original LLaVA-1.5 data. Compared to the original distribution, our synthetic data is more balanced, without categories that are excessively underrepresented.

As shown in Table A4, we retained data solely from five types: VQA (Visual Question Answering), Visual Chat, REG&REC (Region Expression and Recognition), Caption, and Multiple-Choice/Judgement tasks, and tested the performance of VLMs when re-instructed to fine-tune on these filtered data distributions. The results indicate that the VLM generally shows performance improvements in the training sets corresponding to the integrated data types. For instance, the VLM integrated with VQA data outperformed others on the VQAv2 and GQA datasets. However, this trend is not without exceptions. For example, after integrating multiple-choice data, the VLM did not surpass the performance achieved with other data types on the ScienceQA and MM-Vet benchmarks, which are also multiple-choice in nature. Moreover, the integration of Visual Chat data resulted in the highest average performance improvement of 2.3%, while the performance gain from Multiple-Choice/True-False data was the least. This could be due to Visual Chat data containing richer image features compared to other types, thus providing more supervisory information during the fine-tuning phase.

## C.5 Ablation Study on Alternative Masking Ratios for QA pairs

Pilot results suggested limited sensitivity to the exact Q/A masking split, so we used a balanced 50/30/20 (Q-mask/A-mask/None). To verify this, we retrained **Multitask-LLaVA-1.5** with two alternatives: *Q-heavy* (60/20/20) and *A-heavy* (40/40/20).

Table A6: **Comparison with advanced methods and baselines on 8 vision-language benchmarks with error bar. Note:** SRF-LLaVA-1.5 results are reported as mean ± standard deviation over three independent fine-tuning runs with identical hyperparameters and random seeds.

| Method | LLM | Data | LLaVA$^W$ | MM-Vet | MME$^C$ | MMB | MMB$^{CN}$ | VQA$^{v2}$ | GQA | SQA$^I$ |
|---|---|---|---|---|---|---|---|---|---|---|
| LLaVA-1.5 7B | Vicuna-1.5-7B | 665K | 63.4 | 30.5 | 316.1 | 64.3 | 58.3 | 78.5 | 62.0 | 66.8 |
| Recap-LLaVA-1.5 7B | Vicuna-1.5-7B | 665K + 1M | 63.3 | 29.9 | 321.3 | 66.1 | 58.5 | 79.2 | 62.5 | **69.01** |
| **SRF-LLaVA-1.5 7B (mean ± std)** | Vicuna-1.5-7B | 665K + 200K | **67.2 ± 0.62** | **33.1 ± 0.06** | **332.0 ± 0.21** | **66.4 ± 0.16** | **59.1 ± 0.12** | **79.6 ± 0.03** | **62.4 ± 0.25** | 68.2 ± 0.35 |

Then, in Stage 1, generated QA pairs for 1,000 unlabeled images and evaluated their accuracy and diversity: **Acc**↑ (GPT-4o judged image–QA consistency), **TTR**↑ (type–token ratio), and **Distinct$_2$**↑ (unique bigrams ratio). As shown in Table A5, all strategies yield comparable results; we therefore keep $50/30/20$ for simplicity and reproducibility.

Table A5: **Masking configurations and results (Accuracy, Type-Token Ratio, and Distinct-2).**

| Mask Type | Mask Q&A | Mask Q | Mask A | Acc(%) | TTR | Distinct-2 |
|---|---|---|---|---|---|---|
| Original | 50% | 20% | 30% | 85.3 | 0.1144 | 0.4790 |
| More Balanced | 33% | 33% | 33% | 87.7 | 0.0995 | 0.4381 |

## C.6 Error Bars in the Main Experimental Results

We have repeated the entire instruction-fine-tuning phase of SRF-LLaVA three times, each time using identical hyperparameters, the same random seed, and the same training-dataset order. The resulting mean ± standard deviation across the three runs has been reported in Table A6.

# D   More Qualitative Results

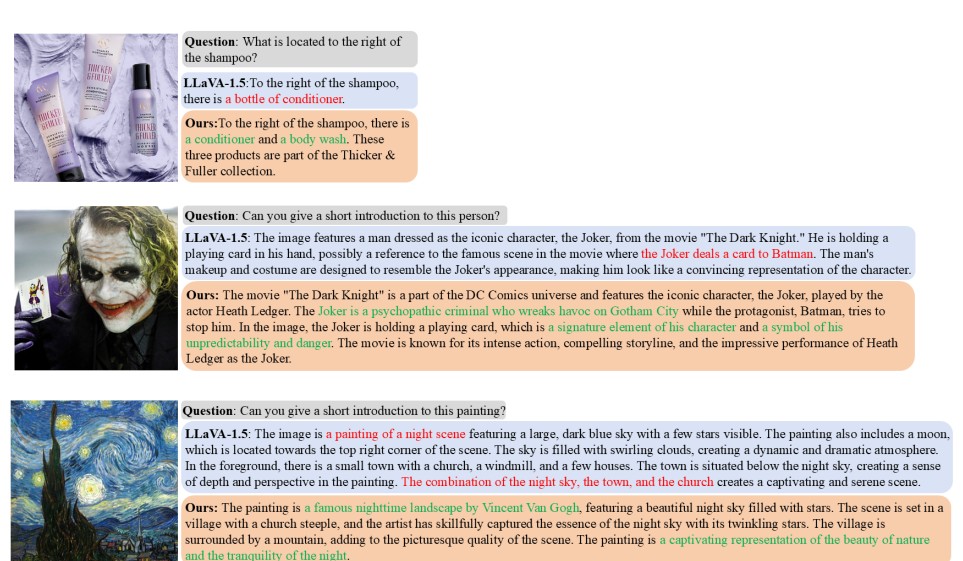

Figure A4: **More qualitative examples of our SRF-LLaVA-1.5 and LLaVA-1.5**. In all three examples, SRF-LLaVA-1.5 provides more detailed and instruction-aligned responses compared to LLaVA-1.5. Red highlights indicate factual errors or irrelevant content in the response, while green highlights emphasize image details critical for providing an accurate answer.

In this section, we present a series of qualitative results to intuitively compare the differences between SRF-LLaVA-1.5 and the original LLaVA-1.5.

**Improvements in Model Capabilities**   From these analyses, we observed several enhancements in our model. As mentioned in Section 4.4, SRF-LLaVA-1.5 exhibits enhanced detail recognition, capturing more subtle features in images than LLaVA-1.5; and possesses richer world knowledge, successfully identifying artists and interpreting underlying meanings in artworks, as shown in

Figure A5: **Failure cases**. Both SRF-LLaVA-1.5 and LLaVA-1.5 fail to correctly understand the implications of the meme. Red highlights indicate factual errors or irrelevant content in the response, while green highlights emphasize image details critical for providing an accurate answer.

Figure A4. In addition to the aforementioned capabilities, we have also observed that SRF-LLaVA-1.5 demonstrates superior instruction-following capabilities, allowing it to flexibly adjust its responses based on different types of prompts. For instance, in the second example of Figure A4, the user's instruction is to introduce the character "Joker." While LLaVA-1.5 continues to focus on describing the image content, our SRF-LLaVA-1.5 provides a comprehensive summary of the Joker as per the user's request, without being confined to the content displayed in the image.

**Failure Case Analysis**   Despite these improvements, SRF-LLaVA-1.5 shares some shortcomings with the original LLaVA-1.5. In a meme presented in Figure A5, which superficially consists of four unrelated images, a deeper analysis reveals concerns about students' situations in online learning. However, both models provide only surface-level analyses of the image. While this limitation is likely due to constraints in the logical reasoning capabilities of the underlying language models used, we are optimistic that future advancements through our Self-Refinement approach will help us overcome this boundary, enabling SRF-LLaVA-1.5 to understand deeper meanings and provide more insightful analyses.

# E   Empirical justification of Equation (4) on synthetic data

To examine the conditions underlying Equation (4), we design a controlled synthetic experiment that follows the generative process in Section 3.3, where the output distribution $P(Y)$ arises from the convolution of a deterministic component with independent noise. Concretely, we draw $X \sim$ Laplace$(0, 1)$ and independent noise $N \sim$ Laplace$(0, 0.6)$ in dimension $d = 50$, then set

$$Y = X\Phi^\top + N,$$

with a random full-rank matrix $\Phi \in \mathbb{R}^{50 \times 50}$. The dataset is split into $N_{\text{lab}} = 1900$ labeled pairs $(X, Y)$ for training, $N_{\text{unl}} = 4900$ unlabeled observations $Y_{\text{new}}$ for self-refinement, and $N_{\text{test}} = 1000$ labeled pairs for evaluation.

A three-layer MLP with two hidden layers of 128 ReLU units takes a test vector $Y \in \mathbb{R}^{50}$ as input and outputs two 50-dimensional vectors $(\mu, b)$, where $\mu$ estimates the conditional median of $X$ and $b \in \mathbb{R}^{50}_{>0}$ the coordinate-wise Laplace scales. The network is trained on labeled data by minimizing the Laplace negative log-likelihood

$$\mathcal{L}(X, \mu, b) = \sum_{j=1}^{d} \left[ \log(2b_j) + \frac{|X_j - \mu_j|}{b_j} \right],$$

using Adam (learning rate $10^{-3}$, batch size 128, 50 epochs). During self-refinement, the model predicts pseudo-labels $\hat{X}_{\text{new}} = \mu(Y_{\text{new}})$, retains the top $40\%$ most confident samples (smallest predicted scales $b$), augments the training set with these pseudo-labels, and retrains; this process is repeated for several rounds. On the test set, we report (i) Laplace NLL as above; (ii) mean squared error

$$\text{MSE} = \frac{1}{N_{\text{test}} d} \sum_{i=1}^{N_{\text{test}}} \|X_i - \mu_i\|_2^2;$$

Table A7: **Synthetic-data self-refinement under the generative model in Section 3.3**. Lower NLL/MSE is better; higher $R^2$ is better.

| Setting | NLL | MSE | $R^2$ |
|---|---|---|---|
| Baseline | 1.1840 | 0.4691 | 0.7635 |
| After refinement | 0.9046 | 0.2872 | 0.8551 |
| Improvement ($\Delta$) | 0.2794 | 0.1819 | 0.0916 |

and (iii) $R^2$ computed by `scikit-learn`'s `r2_score` with `multioutput="uniform_average"` (higher is better). Results are summarized in Table A7. The self-refinement procedure consistently improves NLL, MSE, and $R^2$, supporting the premise that additional observations of the effect variable $Y$ help estimate $P(X \mid Y)$ under the assumed decomposition. We note, however, that directly transferring this decomposition to real-world VLMs remains a theoretical abstraction; see Section 3.3 for discussion of this limitation.

