# OpenReview forum: "Towards Self-Refinement of Vision-Language Models with Triangular Consistency"
_NeurIPS.cc/2025/Conference — NeurIPS 2025 poster_

### Official Review · Reviewer_7zkJ · 2025-06-04

**Clarity:** 2
**Significance:** 2
**Originality:** 2
**Rating:** 4
**Confidence:** 5

**Summary:**

The authors demonstrate that vision–language models (VLMs) can autonomously improve without any external supervision by leveraging a “triangular consistency” among images, questions, and answers: first, the base VLM is instruction-tuned to handle tasks like “image → question-answer” and “image+answer → question,” enabling it to generate candidate triplets from unlabeled images; next, each synthetic triplet is filtered by ensuring that any one element (image, question, or answer) can be accurately reconstructed from the other two, thereby retaining only high-quality pseudo-supervision.

**Questions:**

Refer to Weaknesses

**Ethical Concerns:**

["NO or VERY MINOR ethics concerns only"]

**Final Justification:**

The authors addressed the concerns that initially I mentioned, so I've updated the score. Especially, I recommend the authors should present many variants of models and challenging benchmarks' score as well as clarify ablation studies to be emphasized.

**Limitations:**

yes

**Quality:**

2

**Strengths And Weaknesses:**

Strengths:

- Novel Self-Refinement Framework: Introduces a triangle-consistency principle to activate and guide the self-refinement capability of vision-language models without external supervision.

- Causal-Theoretical Explanation: Provides a clear causal perspective that explains why and how triangle consistency leads to better model performance.


Weaknesses:

I feel it is great work, but there are some four concerns. First one is that the authors only use LLaVA-1.5 as baseline to apply your method, but this paper could be improved by using multiple recent released models such as Qwen2/2.5-VL and InternVL2.5/3 or the models you are using for evaluation baselines. The second is that there are none of challening benchmarks like MM-Vet-v2, MMMU, and MMMU-Pro. Third is that I am wondering if this paper could be improved by the causal theoretical analysis like backdoor adjustment (Treatment T, Outcome Y, Confounder X, and Counterfacture Z). The last one is that there are none of ablation studies regarding your dataset can be used for just SFT or different way training instead of your method. This aims to identify if the performance gain is not just using more dataset or more training time.

---

> ### Author Rebuttal · Authors · 2025-07-31
>
> Dear Reviewer 7zkJ,
>
> We sincerely thank the reviewer for their thoughtful and constructive comments, and for recognizing the novelty of our self-refinement framework and its causal-theoretical grounding. Your comments helped us to improve the completeness of the evaluations of our methods.  In light of your suggestions, we add the evaluation with recent released backbone and new challenging benchmarks and clarify the ablation studies.  We respond to the four main concerns raised below:
>
> > **Q1.** The authors only use LLaVA-1.5 as baseline to apply your method, but this paper could be improved by using multiple recent released models such as Qwen2/2.5-VL and InternVL2.5/3 or the models you are using for evaluation baselines.
>
> **A1.** Thank you for highlighting the importance of demonstrating the generality of our Self‑Refinement (SRF) framework across diverse model families.
> Following your suggestion, we have added Qwen‑2.5‑VL‑3 B as a new baseline. We reused the exact 1 M unlabeled images and three‑stage training pipeline described in the manuscript, and we kept all hyper‑parameters identical to those of the original Qwen 2.5‑VL‑3B release. The resulting model SRF‑Qwen 2.5‑VL‑3B is compared with the vanilla Qwen 2.5‑VL‑3B on 6 widely‑used benchmarks in table below.
>
>
>
> | Model[1]             | MMMU[2] | MMMU Pro[3] | MathVista | MathVision | MMStar | MMBench1.1[4] |
> | ----------------- | ---- | -------- | --------- | ---------- | ------ | ---------- |
> | Qwen2.5-VL-3B     | 53.1 | 31.6     | 62.3      | 21.2       | 55.8   | 81.5       |
> | SRF-Qwen2.5-VL-3B | 55.0 | 31.9     | 64.7      | 23.3       | 56.5   | 80.8       |
>
> [1]: SRF-Qwen 2.5-VL-3B was evaluated with VLMEvalKit; Qwen 2.5-VL-3B scores are taken from the official model documentation.
>
> [2]: Results are obtained on the MMMU validation (VAL) set.
>
> [3]: Results are obtained on the MMMU Pro Standard (10-choice) split.
>
> [4]: Results are obtained on the MMBench1.1 Dev English split.
>
>
> > **Q2.** There are none of challenging benchmarks like MM-Vet-v2, MMMU, and MMMU-Pro.
>
>
> **A2.** We appreciate the recommendations of these challenging benchmarks. In light of your suggestions, we have now evaluated SRF‑LLaVA‑1.5 and the original LLaVA‑1.5 on MMMU and MMMU‑Pro. The new results (see revised Table 1) are shown in the table below.
>
> | Model[5]            | MMMU | MMMU Pro
> | -----------------| ---- | -------- |
> | LLaVA-1.5-7B     | 34.7 | 17.6 |
> | SRF-LLaVA-1.5-7B | 36.6 | 18.3 |
>
> [5]: For both LLaVA-1.5-7B and SRF-LLaVA-1.5-7B, we obtained the results through local evaluation using VLMEvalKit.
>
>
>
>
> > **Q3.** I am wondering if this paper could be improved by the causal theoretical analysis like backdoor adjustment (Treatment T, Outcome Y, Confounder X, and Counterfacture Z).
>
> **A3**: Thank you for the insightful suggestion! We carefully considered the potential of applying causal theoretical frameworks, such as backdoor adjustment, to improve VLMs. Backdoor adjustment is primarily designed to enable causal inference by mitigating bias introduced by observed confounders. However, in our current setting to learn VLMs \hat{X} = f(Y), we do not explicitly identify any clear confounders that would justify applying such an adjustment.
>
> At this stage, it appears that backdoor adjustment may not directly benefit our approach. That said, we are very open to further exploring this direction and would greatly appreciate any additional insights you may have on how such causal tools could be effectively integrated into VLM training or evaluation.
>
>
> > **Q4.** There are none of ablation studies regarding your dataset can be used for just SFT or different way training instead of your method. This aims to identify if the performance gain is not just using more dataset or more training time.
>
>
> **A4.** Thank you for insisting on this clarification. We performed an ablation in which we replaced the Self‑Refinement procedure with a straightforward recapitulation baseline Recap‑LLaVA‑1.5, which was  constructed as follows (The detailed comparison can be found in Table 1:
>
> (i)Starting from the same 1 M unlabeled images used by SRF‑LLaVA‑1.5, we employ the frozen LLaVA‑1.5‑7 B to produce one detailed caption per image.
>
> (ii) We take all 1 M generated captions without any filtering and mix them with the original LLaVA‑1.5 training set.
>
> (iii) We fine‑tune the model under the same schedule as SRF‑LLaVA‑1.5.
>
> Despite having the same data volume and training steps as SRF‑LLaVA‑1.5, Recap‑LLaVA‑1.5 achieves only marginal improvements over the vanilla model (see Table 1). By contrast, SRF‑LLaVA‑1.5 attains substantially larger gains. These results indicate that the benefits stem chiefly from the Triangular Consistency constraint and iterative error‑correction built into Self‑Refinement, rather than from simply exposing the model to more unlabeled image data or longer optimization.
>
> Additionally, the ablation studies presented in Table 3 (the “Consistency” group) further support that the performance gains are not solely due to the use of unlabeled data. In particular, the “Consistency” Bottom 20% baseline applies the same self-refinement procedure but selects self-generated image-text pairs with low consistency scores. Although this baseline also leverages the same unlabeled data, it lacks our consistency-based filtering mechanism. The notable accuracy gap between our model and this baseline highlights that the improvement is primarily driven by consistency-guided self-refinement, rather than merely the presence of additional data.

---

### Official Review · Reviewer_zL5Z · 2025-06-12

**Clarity:** 2
**Significance:** 2
**Originality:** 3
**Rating:** 4
**Confidence:** 4

**Summary:**

This work investigates the self-refinement capabilities of VLMs by introducing a training scheme where the model is trained to learn not only how to generate an answer (A) given an image (I) and a question (Q), but also how to generate A given (I, Q) and Q given (I, A). By forcing the model to learn these extra tasks, it is then used to generate IQA triplets for unlabeled images. The generated instructions are then filtered by the proposed triangular consistency check: for each synthetic triplet, the model tries to regenerate the question from the I+A and the answer from the I+Q, and the results are compared to the original Q and A. The model is then fine-tuned on the filtered synthetic data, improving its capabilities without any human annotations. Empirical results indicate that the method yields consistent yet modest improvements.

**Questions:**

- In line 201, what is the rationale for this relation $Y = \phi (X, N_Y)$? I read [27], but I cannot see how you can explicitly use their assumption to VLMs.
- Is there any empirical justification for (4) specifically for VLMs?
- Given that SRF-LLaVA-1.5 has an unfair advantage over vanilla LLaVA-1.5 since the former has been expose to more images (even though unlabeled), have you tried to check the performance if you just use the same extra images with their true (Q, A) pairs (assuming there are available) to check how close you are to the fully supervised model. That could be considered as an upper bound for SRF.

**Ethical Concerns:**

["NO or VERY MINOR ethics concerns only"]

**Final Justification:**

My concerns have been addressed in the author's responses, so I will keep my initial score. Incorporating this discussion in the final version would strengthen the readability of the paper.

**Limitations:**

Yes, but the discussion about limitations is too short.

**Paper Formatting Concerns:**

There are no major formatting issues.

**Quality:**

2

**Strengths And Weaknesses:**

# Strengths
- The proposed three-step framework is logically sound and practical to implement.
- The authors conduct extensive experiments on eight benchmarks, with results consistently favouring the self-refined model over the baseline, demonstrating the robustness of the improvement.


# Weaknesses
- Despite the consistent improvement of SRF-LLaVA over the vanilla version, most of the accuracy gaps are narrow, and there are no error bars to justify any statistical significance.
- There is no total running time comparison between SRF-LLava and SRF-LLaVA. In general, SRF-LLava would require more computational resources, and it is not clear to a practitioner what is the computational cost is required to gain a few percentages of accuracy improvement.
- The theoretical analysis is based on several arbitraty assumtions that is hard to verify and there is no discussion why the authors resort to these assumptions.

---

> ### Author Rebuttal · Authors · 2025-07-31
>
> Dear Reviewer zL5Z,
>
> We sincerely thank the reviewer for their thoughtful evaluation and recognition of our framework’s soundness, practicality, and consistent improvements across eight benchmarks. We appreciate the opportunity to clarify several points and address the concerns raised.
>
> > **Q1.** Despite the consistent improvement of SRF-LLaVA over the vanilla version, most of the accuracy gaps are narrow, and there are no error bars to justify any statistical significance.
>
> **A1.** Thank you for pointing this out. While the accuracy gaps are sometimes narrow, we respectfully emphasize that the consistent improvements of SRF‑LLaVA are achieved under the same training configurations as LLaVA, without relying on any additional human annotations or more powerful VLMs such as GPT‑4V.
>
> In line with your advice, we have repeated the entire instruction-fine-tuning phase of SRF-LLaVA three times, each time using identical hyperparameters, the same random seed, and the same training-dataset order. The resulting mean ± standard deviation across the three runs has been reported in the updated Table 1 of the main paper, as shown in the table below.
>
>
>
> | Model                         | LLaVA-Wild  | MM-Vet      | MME-C        | MMBench     | MMBench-CN  | VQAv2       | GQA         | ScienceQA   |
> | ----------------------------- | ----------- | ----------- | ------------ | ----------- | ----------- | ----------- | ----------- | ----------- |
> | LLaVA-1.5-7B                  | 63.4        | 30.5        | 316.1        | 64.3        | 58.3        | 78.5        | 62.0        | 66.8        |
> | SRF-LLaVA-1.5-7B (mean ± std) | 67.2 ± 0.62 | 33.1 ± 0.06 | 332.0 ± 0.21 | 66.4 ± 0.16 | 59.1 ± 0.12 | 79.6 ± 0.03 | 62.4 ± 0.25 | 68.2 ± 0.35 |
>
>
>
>
>
>
>
>
>
> > **Q2.** There is no total running time comparison between SRF-LLaVA and LLaVA. In general, SRF-LLava would require more computational resources, and it is not clear to a practitioner what is the computational cost is required to gain a few percentages of accuracy improvement.
>
> **A2.** We regret the omission and have added full training‑time statistics to Section 4.4 in the revised version. At inference time, SRF‑LLaVA‑1.5 and LLaVA‑1.5 are identical in parameter count and therefore require exactly the same FLOPs and GPU memory.
>
> During the training of SRF-LLaVA-1.5, the larger number of images in the dataset naturally led to a longer training time. The table below compares the training durations of SRF-LLaVA-1.5 with those of the baseline LLaVA-1.5, both trained on a cluster equipped with 8 NVIDIA H100-NVL GPUs (96 GB each).
>
>
> | Model | Total wall‑clock time | Effective GPU‑hours |
> | -------- | -------- | -------- |
> | LLaVA‑1.5     | 7 h     | 56     |
> | SRF‑LLaVA‑1.5  | 12 h     | 96     |
>
> Thus the refinement procedure costs an additional 40 GPU‑hours (≈0.7×). We now state this trade‑off explicitly in the paper so that practitioners can make an informed decision.
>
>
> >**Q3.** The theoretical analysis is based on several arbitraty assumtions that is hard to verify and there is no discussion why the authors resort to these assumptions.
>
>
> **A3.** Thanks for the questions. These assumptions we employed include (1) the independence of mechanism and input distribution, and (2) the Additive Noise Model (ANM).
> These assumptions are well-established functional assumptions in causal inference literature, such as [1,2,3] rely on ANM and the independence of mechanism and input distribution to identify the causal directions.
>
> 1. Why do we adopt these assumptions?
>
> (1) The independence of mechanism and input indicates that P(Y; image|X; text) contains no information about P(X; text). It introduces an asymmetry between cause and effect, since it will usually be violated in the backward direction. In Section 3.1, we discussed why this asymmetry holds true. Empirically, this asymmetry aims to provide implications about VLMs that we can use unlabeled images to produce the self-refinement of VLMs, but it is hard to use unpaired text sentences to produce the self-refinement.
>
> (2) The ANM, especially $Y = φ(X) + N_Y$, provides a functional assumption of the model in which the causal direction can be proved for identification. With such an assumption, the causal directions can be verified.
>
> 2. Verifiability.
> While these assumptions may not be directly verifiable in the context of vision-language models (VLMs), they serve as necessary theoretical tools to justify plausible structural properties and to derive interpretable insights. These simplifying yet principled assumptions allow us to provide a theoretical explanation for why self-refinement is achievable.
>
> In response to your comment, we have added a discussion in Section 3.3 of the revised manuscript to explicitly motivate the choice of these assumptions, clarify their connection to prior foundational work, and acknowledge their limitations.
>
> Please let us know if any part is unclear. We are more than happy for a deeper discussion.
>
>
> [1] Zhang, Kun, and Aapo Hyvarinen. On the identifiability of the post-nonlinear causal model. 2012.
> [2] Shohei Shimizu, Patrik O Hoyer, Aapo Hyv¨arinen, and Antti Kerminen. A linear non-Gaussian acyclic model for causal discovery. 2006.
> [3] Janzing, D. and Scholkopf, B. Causal inference using the algorithmic Markov condition.  2010.
>
> > **Q4.** In line 201, what is the rationale for this relation $Y=\phi(X, N_Y)$? I read [27], but I cannot see how you can explicitly use their assumption to VLMs.
>
> **A4.** Thank you for the question. The equation $Y = \phi(X, N_Y)$ is intended to model the data generation process, not the behavior of the trained system. It follows the functional causal model framework, where text X is treated as the cause and images Y as the effect, which is consistent with the causal direction illustrated in Figure 3.
>
> We do not impose this assumption during VLM training. Instead, we adopt this formulation to theoretically motivate our self-refinement approach. The key intuition is that unlabeled image data enables a more accurate estimation of the marginal distribution P(Y) (i.e., of images). Under the assumptions of additive noise and causal sufficiency, this marginal can reveal structure useful for learning a VLM that predicts text from images,i.e., modeling $\hat{X} = f(Y)$.
>
>
> > **Q5.** Is there any empirical justification for (4) specifically for VLMs?
>
> **A5.** Thank you for the question! Regarding Equation (4), we currently provide empirical justification only in the context of synthetic data. Specifically, we construct data that follows the generative process outlined in Section 3.3, where the output distribution P(Y) is formed via a convolution of an underlying deterministic component and an independent noise variable.
>
>
> We draw $X\sim\mathrm{Laplace}(0,1)$ and independent noise $N\sim\mathrm{Laplace}(0,0.6)$ in dimension $d=50$, then set $Y=X\Phi^{\top}+N$ with a random full-rank $\Phi\in\mathbb{R}^{50\times50}$. The data are split into $N_{\text{lab}}=1900$ labeled pairs $(X,Y)$ for training, $N_{\text{unl}}=4900$ unlabeled observations $Y_{\text{new}}$ for self-refinement,  and $N_{\text{test}}=1000$ labeled pairs for evaluation. A three-layer MLP with two hidden layers of 128 ReLU units takes a test vector $Y\in\mathbb{R}^{50}$ as input and outputs two 50-dimensional vectors $(\mu,b)$: $\mu$ estimates the conditional median of $X$ and $b>0$ its coordinate-wise Laplace scale. The network is trained on labeled data with the Laplace negative-log-likelihood loss $\mathcal{L}(X,\mu,b)=\log(2b)+|X-\mu|/b$ using Adam (learning rate $10^{-3}$, batch 128, 50 epochs). During self-refinement, the model predicts pseudo-labels $\hat X_{\text{new}}=\mu(Y_{\text{new}})$, keeps the 40 % most confident samples (smallest predicted scales $b$), retrains on the augmented set, and repeats for several rounds. On the test set we report **MSE**: $=\frac{1}{N_{\text{test}}\!d}\sum_{i=1}^{N_{\text{test}}}\lVert X_i-\mu_i\rVert_2^{2}$  and **$R^{2}$** computed by scikit-learn’s `r2_score` with `multioutput="uniform_average"` (higher is better).
> | Setting            | NLL   | MSE    | $R^{2}$ |
> |--------------------|:-----:|:------:|:-------:|
> | Baseline           | 1.1840 | 0.4691 | 0.7635 |
> | After refinement   | 0.9046 | 0.2872 | 0.8551 |
> | Improvement        | $\Delta=0.2794$ | $\Delta=0.1819$ | $\Delta=0.0916$ |
>
>
> This controlled setup allows us to empirically verify the assumed conditions, and we can use more data of the effect variable Y to help the estimation of P(X|Y). However, we acknowledge that applying this decomposition directly to real-world VLMs remains a theoretical abstraction. We have clarified this limitation in Section 3.3 of the revised manuscript.
>
> > **Q6.** Given that SRF-LLaVA-1.5 has an unfair advantage over vanilla LLaVA-1.5 since the former has been expose to more images (even though unlabeled), have you tried to check the performance if you just use the same extra images with their true (Q, A) pairs (assuming there are available) to check how close you are to the fully supervised model. That could be considered as an upper bound for SRF.
>
> **A6.** We appreciate the constructive suggestion! We totally agree that it is an ideal manner to give an upper bound for SRF. Unfortunately, the extra images come from LAION and are unlabeled. Obtaining automated annotation with state‑of‑the‑art VLMs is beyond our budget: using GPT‑4V api to produce detailed captions for 2.8 M images would cost roughly $19,152 (estimated from the current OpenAI pricing).
> We therefore apply an alternative manner to use open source sub-optimal model, Qwen 2.5-VL-32B to annotate the (Q, A) pairs.However, because of the model’s limited inference speed, the image annotations are still underway; we will update the experimental results during the discussion phase.

---

> > ### Comment · Reviewer_zL5Z · 2025-08-05
> > **Response**
> >
> > Thank you for your rebuttal. I appreciate the time you have taken to address my concerns. I am satisfied with your response, so I will keep my initial score. I believe incorporating this discussion in the final version would help a lot with the clarity and readability of your work.

---

> > > ### Author Response · Authors · 2025-08-08
> > > **We highly appreciate your supports and valuable inputs which improved our paper a lot!**
> > >
> > > Dear Reviewer zL5Z,
> > >
> > > We sincerely appreciate your time and the valuable suggestions that have helped improve the clarity of our paper. All discussions have been incorporated into the final version. As the rebuttal discussion period draws to a close, we would like to confirm whether your concerns have been fully addressed. We would be more than happy to provide further clarifications or justifications before the discussion ends if needed.
> > >
> > > If you find the revisions satisfactory, we would be grateful if you might consider updating your rating accordingly. Thank you again for your thoughtful feedback!
> > >
> > > Best regards,
> > >
> > > The authors

---

> ### Author Response · Authors · 2025-08-05
> **Thanks for your feedback!  We’re pleased that you are satisfied with the response and keep positive score.**
>
> Thank you for your feedback, and we sincerely appreciate your suggestions, which have helped improve the clarity and readability of the paper. We are pleased that our responses have addressed your concerns and will incorporate these discussion points into the final version.

---

### Official Review · Reviewer_rr1J · 2025-06-29

**Clarity:** 2
**Significance:** 3
**Originality:** 2
**Rating:** 4
**Confidence:** 4

**Summary:**

This work introduced a metric based on Vision-Question-Answer data to measure the reliability of generated instructions on unlabeled data. This work presented Self-Refinement Framework for data generation and provided theoretical analysis for the framework.

**Questions:**

1. Was the dataset used in the MobileVLM model generated by itself or LLaVA-1.5?
2. How do you tell the model to generate specific type of data as the prompts shown in Figure A1 do not provide the data type information?
3. The link shown in line 100 of Appendix is expired.
4. What is the meaning of "k" and "k+1" shown in Figure2? Please provide more explanations about the overview of the framework.

**Ethical Concerns:**

["NO or VERY MINOR ethics concerns only"]

**Final Justification:**

After reading the answers, my questions about the methods and content have been resolved, so I will keep my original score. I hope the author can address these concerns in the final version of the paper.

**Limitations:**

yes

**Quality:**

3

**Strengths And Weaknesses:**

Strengths:
1. This paper introduced a self-refinement framework to utilize the capacity of the pre-trained VLM in data generation.
2. This work conducted comprehensive experiments in ablation study for different types 他the generation data.

Weaknesses:
1. It is not fair for comparison with baseline as the SRF-LLaVA-1.5 is trained on a larger dataset than as shown in Table 1. And the paper does not provide sufficient details about used data and training procedures about Recap-LLaVA-1.5.
2. There seems to be a conflict between the 2.8 million images, mention in line 227, and the external knowledge, mention in line 73

---

> ### Author Rebuttal · Authors · 2025-07-31
>
> Dear Reviewer rr1J,
>
> We are deeply grateful for your positive assessment of our ablation-study design and for the valuable time you devoted to a thorough review of our manuscript. Your insightful remarks regarding the experimental workflow, baseline settings, and data-generation details have provided us with an excellent opportunity to enhance the paper’s clarity and readability. In accordance with your suggestions, we have added a more comprehensive discussion of the model-training pipeline and the data-generation process.
>
>
> > **Q1.** It is not fair for comparison with baseline as the SRF-LLaVA-1.5 is trained on a larger dataset than as shown in Table 1. And the paper does not provide sufficient details about used data and training procedures about Recap-LLaVA-1.5.
>
>
> **A1.** Thank you for raising this concern. SRF‑LLaVA‑1.5 is trained on an expanded dataset that includes an additional 200K unlabeled images compared to the original LLaVA. Thus, to ensure a fair comparison, we included evaluations against more baseline models that are also trained with the same augmented dataset.
>
> Specifically, we compare SRF‑LLaVA‑1.5 with Recap‑LLaVA‑1.5 (Table 1), and the “Consistency” Bottom 20% model (4th row in Table 3), both of which use the additional unlabeled images. These comparisons demonstrate that the performance gains primarily stem from our self-refinement process, rather than simply the inclusion of more data.
>
> To isolate the effect of self-refinement, we construct Recap‑LLaVA‑1.5 using the following two-stage procedure:
> 1. Use LLaVA‑1.5 (7B) to caption 1M unlabeled images. (We call this process Recap.)
> 2. Combine the resulting self-captioned image-text pairs with the original LLaVA‑1.5 training set, and finetune using the same training configurations as LLaVA‑1.5 and SRF‑LLaVA‑1.5.
> Although Recap‑LLaVA‑1.5 sees the same enlarged corpus, its performance gains are significantly smaller than those of SRF‑LLaVA‑1.5, as shown in Table 1.
>
> Similarly, the “Consistency” Bottom 20% baseline follows the same self-refinement procedure but selects self-generated image-text pairs with lower consistency scores. While this model also leverages the unlabeled data, it does not apply our consistency-based filtering. The accuracy gap between our model and this baseline further supports that the improvement comes from our consistency-guided self-refinement, not just the volume of unlabeled data.
>
> In light of your suggestion, we have added the training details of Recap‑LLaVA‑1.5 to Section 4.3 of the revised manuscript.
>
>
> > **Q2.** There seems to be a conflict between the 2.8 million images, mention in line 227, and the external knowledge, mention in line 73
>
> **A2.** We apologize for the unclear wording and thank you for pointing it out. In line 73, we meant “no external annotations were used.” The model is trained only on image–text pairs that are self-produced by the model itself; No additional human-labeled data or significantly stronger VLMs, such as GPT-4V, were employed. In light of your suggestions, we have replaced "external knowledge" with "external annotations by human or other stronger VLMs" in the revised version.
>
>
> > **Q3.** Was the dataset used in the MobileVLM model generated by itself or LLaVA-1.5?
>
> **A3.** Thank you for this question. The merged dataset for MobileVLM consists of (i) samples generated by MobileVLM itself through our self‑refinement loop on the 1 M unlabeled images, and (ii) its original supervised training set. We have added a detailed breakdown and training recipe in the appendix.
>
> > **Q4.** How do you tell the model to generate specific type of data as the prompts shown in Figure A1 do not provide the data type information?
>
> **A4.** We highly appreciate your insightful question. We do not enforce any particular question–answer type. The prompts simply ask the model to generate a question and its answer. The distribution of generated QA types is therefore emergent; Fig. A3 reports those proportions. We have clarified this in the caption of Fig. A1 in the revised version.
>
>
> > **Q5.** The link shown in line 100 of Appendix is expired.
>
> **A5.** Thank you for pointing it out. Because the NeurIPS 2025 rebuttal rules prohibit updating anonymous repositories during the discussion phase, the link must remain frozen for now. We will refresh the repository and verify permanent access immediately after the rebuttal period.
>
>
> > **Q6.** What is the meaning of "k" and "k+1" shown in Figure2? Please provide more explanations about the overview of the framework.
>
> **A6.** Thanks for your question. In Fig. 2, k denotes the current self‑refinement iteration and k + 1 the next. For example, the initial model LLaVA‑1.5 is M(0); after one self‑refinement round, it becomes SRF‑LLaVA‑1.5, labeled M(1). M(1) can seed another round, yielding M(2), and so on. We have added an explanatory sentence to the caption and extended the main‑text overview in the updated version for clarity. Below is our core algorithm (presented in pseudocode):
> ```
> Input:
>     M0 ... initial VLM
>     D0 ... human‑labeled (I,Q,A) dataset
>     U  ... unlabeled images
>     K  ... number of refinement rounds
>
>
> for k = 1 … K:	# ----- iterative self‑refinement -----
>
>     # Stage 1 – Multi‑task fine‑tune (caption, VQA, instruction)
>     M0_gen ← train M0 on D0
>     S ← { (I,Q,A) produced by M0_gen for every I ∈ U }
>     S'← { (I,Q',A') | Q' ← M0_gen(I,A) ∧ A' ← M0_gen(I,Q)}
>
>     # Stage 2 – Generate & filter synthetic IQA
>     For (I,Q,A) ∈ S, (I,Q',A') ∈ S':
>     	if Q == Q' and A == A':
>     		F ← (I,Q,A)
>
>     # Stage 3 – Instruction tuning with filtered data
>     D1 ← D0 ∪ F
>     M1 ← train M0 on D1  (instruction‑only objective)
>
> return M
> ```

---

> > ### Author Response · Authors · 2025-08-04
> > **Could you please let us know whether our responses and updated submission properly addressed your concern?**
> >
> > Dear reviewer rr1J,
> >
> > Thank you for dedicating your time to reviewing our work and for your insightful feedback. As the discussion window draws to a close, we wish to confirm whether our broadened explanation of the model-training pipeline and the data-generation workflow fully addresses your concerns. If any further clarification would be helpful, please do not hesitate to let us know. We would be pleased to elaborate.

---

> > > ### Comment · Reviewer_rr1J · 2025-08-06
> > > **Response for Rebuttal**
> > >
> > > Thank you for the answers. After reading the answers, my questions have been partially resolved, so I will keep my original score. I hope the author can address these concerns in the final version of the paper.

---

> > > > ### Author Response · Authors · 2025-08-06
> > > > **Thank you for your feedback and keeping your positive score.**
> > > >
> > > > Thank you for your feedback! We greatly appreciate your suggestions, which have helped improve the paper’s readability. We will incorporate these points into the final version. Thanks again for your valuable input.

---

> ### Author Response · Authors · 2025-08-08
> **Thanks for the valuable suggestions and questions.**
>
> Dear Reviewer rr1J,
>
> We sincerely appreciate your valuable suggestions and questions, which have helped us improve the clarity and quality of our paper. As the rebuttal discussion period is coming to a close, we would like to confirm whether all your concerns have been addressed. If any points remain unclear, please kindly let us know. We will be more than happy to provide additional clarifications or justifications before the end of the rebuttal discussion.
>
> If you feel our work merits it, we would be grateful if you might consider updating your rating accordingly. Thank you again for your thoughtful and invaluable feedback!
>
> Best regards,
>
> The authors

---

### Official Review · Reviewer_6Atd · 2025-06-29

**Clarity:** 3
**Significance:** 2
**Originality:** 2
**Rating:** 4
**Confidence:** 4

**Summary:**

In this paper, the authors leverage the inherent self-refinement capabilities of VLMs to generate high-quality supervised instruction data without external supervision. To achieve this, the authors propose a triangular consistency principle within the image-question-answer triplets: any masked elements should be consistently and accurately reconstructed. Based on this principle, the authors construct a synthetic instruction tuning dataset to further fine-tune the VLMs. Experimental results on standard multimodal benchmarks demonstrate the effectiveness of the proposed method.

**Questions:**

See the questions mentioned above. Given the current status of the paper, I am leaning towards borderline reject and hope the authors could address my concerns during the rebuttal.

**Ethical Concerns:**

["NO or VERY MINOR ethics concerns only"]

**Final Justification:**

The rebuttal well addressed my concerns. After carefully reading other reviewers' comments, I would like to increase my score to borderline accept. I suggest the authors incorporate all the results/comments discussed in the rebuttal into the final version.

**Limitations:**

The authors briefly discussed the limitations in Sec. 5. However, I cannot find a specific section to discuss the societal impacts of their work. Some broader impacts that come to my mind: positive impacts: reduce human supervision and costs, negative impacts: induce potential bias using the synthetic data from the model.

**Paper Formatting Concerns:**

I do not have any formatting concerns.

**Quality:**

3

**Strengths And Weaknesses:**

**Strengths:**

-	The idea of using internal feedback from the model itself to enhance VLMs is interesting.
-	The paper is generally well-written and easy to follow.
-	The experiments are extensive and the results are promising, while some parts need to be improved (see weaknesses).

**Weaknesses:**

-	The authors mainly focus on LLaVA. What is the generalizability of the proposed method to other kinds of VLMs (e.g., Qwen-VL, DeepSeek-VL, Intern-VL)?
-	For LLaVA, the authors mainly focus on 7B and 1.7B models. What is the scalability of the proposed method for larger models (e.g., 13B)?
-	According to L237-238, the authors use different masking ratios for Q and A. How to decide the ratios? Using different ratios can potentially influence the final performance. However, it seems that no relevant ablations are conducted.
-	Some details are missing exact descriptions. When the authors fine-tune the VLM on the merged dataset, it is unclear which specific modules are fine-tuned (the whole model or some sub-modules)? And what is the effect of fine-tuning different modules?
-	Although the authors mention that the self-refinement process can be conducted continuously, it seems that the performance has already saturated after one-round update. This makes the potential of the proposed method somewhat limited.

---

> ### Author Rebuttal · Authors · 2025-07-31
>
> Dear Reviewer 6Atd,
> We sincerely thank you for recognizing the originality of our work, for your encouraging comments on our experimental design and results, and for the valuable suggestions you offered to further enrich our study. We are also deeply grateful for the time and effort you devoted to reviewing our manuscript.
>
> In response to your recommendations, we have conducted additional extension and ablation experiments and have expanded the description of our implementation details. Our point-by-point replies are presented below:
>
> > **Q1.** The authors mainly focus on LLaVA. What is the generalizability of the proposed method to other kinds of VLMs (e.g., Qwen-VL, DeepSeek-VL, Intern-VL)?
>
> **A1.** Thank you for raising this question regarding the generalizability of our method across different VLMs. To demonstrate that our self-refinement framework (SRF) is not limited to LLaVA, we validated its effectiveness on MobileVLM, as discussed in Section 4.5. In light of your suggestion, we further evaluated the SRF framework using the Qwen 2.5-VL-3B model, one of the strongest open-source VLMs of similar scale. Specifically, we applied our entire pipeline using the same 1 million unlabeled images. The results, summarized in the table below, confirm that our method maintains its effectiveness across different VLM architectures.
>
>
> | Model[1]             | MMMU[2] | MMMU Pro[3]  | MathVision | MMStar | MMBench1.1[4] |
> | ----------------- | ---- | -------- | ---------- | ------ | ---------- |
> | Qwen2.5-VL-3B     | 53.1 | 31.6     | 21.2       | 55.8   | 81.5       |
> | SRF-Qwen2.5-VL-3B | 55.0 | 31.9     | 23.3       | 56.5   | 80.8       |
>
> [1]: SRF-Qwen 2.5-VL-3B was evaluated with VLMEvalKit; Qwen 2.5-VL-3B scores are taken from the official model documentation.
> [2]: Results are obtained on the MMMU validation (VAL) set.
> [3]: Results are obtained on the MMMU Pro Standard (10-choice) split.
>
> [4]: Results are obtained on the MMBench Dev English split.
>
> To provide a clear, side-by-side comparison of model performance, we have added the evaluation results of LLaVA-1.5 7B and SRF-LLaVA-1.5 7B on MMMU and MMMU-Pro, as shown in the table below.
>
> | Model[5]            | MMMU | MMMU Pro
> | -----------------| ---- | -------- |
> | LLaVA-1.5-7B     | 34.7 | 17.6 |
> | SRF-LLaVA-1.5-7B | 36.6 | 18.3 |
>
> [5]: For both LLaVA-1.5-7B and SRF-LLaVA-1.5-7B, we obtained the results through local evaluation using VLMEvalKit.
>
>
> > **Q2.** For LLaVA, the authors mainly focus on 7B and 1.7B models. What is the scalability of the proposed method for larger models (e.g., 13B)?
>
> **A2.** We appreciate the suggestion to conduct experiments on larger backbones. Following your suggestion, we applied our self-refinement framework to LLaVA‑1.5‑13B. We evaluated SRF-LLaVA-1.5-13B on eight standard benchmarks. As illustrated in the following table, our approach yields consistent performance gains over the original LLaVA-1.5-13B across all tasks. These updated results have been incorporated into Table 1 of the revised manuscript.
>
>
> | Model | LLaVA-Wild | MM-Vet | MME-C | MMBench | MMBench-CN | VQAv2 | GQA | ScienceQA |
> |------|--------|--------|------|-----|-------|-------|-----|------|
> | LLaVA-1.5-13B | 70.7 | 35.4 | 295.4 | 67.7 | 63.6 | 80.0 | 63.3 | 71.6 |
> | SRF-LLaVA-1.5-13B | 73.5 | 37.7 | 334.2 | 68.6 | 64.0 | 81.3 | 65.1 | 72.2 |
>
>
> > **Q3.** According to L237-238, the authors use different masking ratios for Q and A. How to decide the ratios? Using different ratios can potentially influence the final performance. However, it seems that no relevant ablations are conducted.
>
> **A3.** Thank you for highlighting this point. In our pilot experiments, we found that model performance was largely insensitive to the exact masking ratios applied to questions and answers. Consequently, we adopted a balanced split of 50 %, 30 %, and 20 % chosen at random.
>
> To address your concern, we have now added an ablation study on alternative masking ratios (i.e., different Stage-1 training-data partitions) in Appendix C of the revised manuscript. Specifically, we retrained Multitask-LLaVA-1.5 under two distinct masking configurations. In the first stage of the Self-Refinement framework, this model generates QA pairs from unlabeled images. We then used each retrained version to create QA pairs for 1,000 unlabeled images and evaluated their diversity and accuracy with GPT-4o. As the table below shows, both masking strategies delivered comparable performance. The Acc metric assesses how well the generated question-answer pairs align with the image content, whereas TTR and Distinct2 measure the lexical diversity present in those QA pairs. (all higher is better)
>
> | Mask Type     | Mask Q&A  | Mask Q    | Mask A    | Acc  | TTR    | Distint2 |
> | ------------- | ---- | ---- | ---- | ---- | ------ | -------- |
> | Original      | 50%  | 20%  | 30%  | 85.3 | 0.1144 | 0.4790   |
> | More Balanced | 33%  | 33%  | 33%  | 87.7 | 0.0995 | 0.4381   |
>
> If you would like us to include results for additional partitions, please let us know—we would be more than happy to add them.
>
>
> > **Q4.** Some details are missing exact descriptions. When the authors fine-tune the VLM on the merged dataset, it is unclear which specific modules are fine-tuned (the whole model or some sub-modules)? And what is the effect of fine-tuning different modules?
>
> **A4.** Thank you for this question, which helped us to clarify the training details.  During fine‑tuning on the merged dataset, we follow the standard LLaVA‑1.5 instruction‑tuning recipe. Specifically, we keep the vision encoder frozen and fine-tune both the image‑text projection layer and the LLM parameters.
> Empirically, updating only the projection layer yields a severe drop in multi‑turn reasoning and fails to correct linguistic hallucinations. Conversely, updating only the LLM impairs pure‑text tasks because the LLM over‑fits to visual cues. Jointly tuning both components achieves the best trade‑off. We have added these implementation details in Section 4.2 of the revised manuscript.
>
>
> > **Q5.** Although the authors mention that the self-refinement process can be conducted continuously, it seems that the performance has already saturated after one-round update. This makes the potential of the proposed method somewhat limited.
>
>
> **A5.** Thank you for this insightful observation. We acknowledge that in our experiments, the performance improvements tend to plateau after the first round of self-refinement. To provide clearer guidance to readers, we explicitly acknowledged the experimental findings in Section 4.5 and the limitations section. In light of your suggestions, we have added more discussions about this. We believe the diminishing returns may be due to the finite amount of useful information that unpaired images can provide about the underlying image distribution. Once this information is sufficiently leveraged, further gains naturally converge. At the same time, we observed that a second round of self-refinement still brings measurable improvements, suggesting that the full potential of this information cannot be fully exploited in just one iteration.
>
>
>
> > **Q6.** The authors briefly discussed the limitations in Sec. 5. However, I cannot find a specific section to discuss the societal impacts of their work. Some broader impacts that come to my mind: positive impacts: reduce human supervision and costs, negative impacts: induce potential bias using the synthetic data from the model.
>
> **A6.** We greatly appreciate your suggestion to include a dedicated discussion on the societal impacts of our work. We fully agree with the positive and negative aspects you mentioned. In light of your suggestions, we have added the following section about social impact:
>
> By demonstrating that VLMs can autonomously improve themselves using only unlabeled images, our approach significantly reduces the reliance on expensive and labor-intensive human annotations. This has the potential to democratize access to high-performance VLMs, especially for low-resource domains and communities with limited labeled data. Despite the benefits, the use of synthetic data generated by the model itself introduces a risk of reinforcing or amplifying existing biases present in the pretrained VLM. Since no human-in-the-loop supervision is involved during the self-refinement process, biases or hallucinations in the model’s generation may propagate or compound. Furthermore, the autonomous nature of the learning pipeline could lead to unintended model behaviors that are harder to trace or audit.

---

> ### Author Response · Authors · 2025-08-04
> **Could you kindly let us know if our recent revisions and expanded clarifications have fully resolved your concerns?**
>
> Dear reviewer 6Atd,
>
> Thank you very much for your time and thoughtful review. As the discussion period is approaching its conclusion, we would be grateful to know whether our additional extension and ablation experiments, together with the expanded description of our implementation details, have satisfactorily addressed your concerns. We appreciate the oppotunity to provide any further explanation if you think it is needed.

---

> > ### Comment · Reviewer_6Atd · 2025-08-06
> > **Response to Rebuttal**
> >
> > Thanks for the authors' rebuttal. The rebuttal well addressed my concerns. After carefully reading other reviewers' comments, I would like to increase my score. I suggest the authors incorporate all the results/comments discussed in the rebuttal into the final version.

---

> > > ### Author Response · Authors · 2025-08-06
> > > **Thank you for your comments and raised score; we are pleased your concerns are well addressed.**
> > >
> > > Thank you for your feedback and for increasing your score after reviewing our rebuttal.  We are delighted that our responses have addressed your concerns and grateful for your constructive suggestions. We will incorporate every additional result and clarification discussed into the final manuscript. Your support is greatly appreciated as we continue refining the paper.

---

### Note · Authors · 2025-08-12

Dear AC and Reviewers

We thank the AC for your time and contributions to the community. We thank the reviewers for their time, insightful suggestions, and kind recognition of our contribution. We are pleased that all reviewers found our response helpful, and would like to raise scores or maintain positive recommendations. We have incorporated all results and discussions into the final version and summarize them below.

**To Reviewer 6Atd (Q1) and Reviewer 7zkJ (Q1)**, we have added experiments on the new baseline Qwen-2.5-VL-3B in Table 2.

**To Reviewer 6Atd (Q2)**, we have added results in Table 1 for the larger-parameter baseline LLaVA-1.5-13B.

**To Reviewer 6Atd (Q3)**, we have added an ablation study in Appendix C.4 analyzing how mask ratios for Q and A affect the quality of QA pairs.

**To Reviewer 6Atd (Q4)**, we have provided fine-tuning details in Section 4.2.

**To Reviewer 6Atd (Q5)**, we have added analysis in Section 4.5 explaining why performance gains slow after the first iteration.

**To Reviewer 6Atd (Q6)**, we have expanded Section 5 to discuss the societal impact.

**To Reviewer rr1J (Q1) and Reviewer 7zkJ (Q4)**, we have extended the description of the training process and data for Recap-LLaVA-1.5 in Section 4.3.

**To Reviewer rr1J (Q2)**, we have replaced “external knowledge” with “external annotations by humans or other stronger VLMs” to make it more clear.

**To Reviewer rr1J (Q3)**, we have added the training details of MobileVLM in Appendix B.5.

**To Reviewer rr1J (Q4)**, we provided the explanation of QA format.

**To Reviewer rr1J (Q5)**, we will update the repository immediately after the rebuttal period.

**To Reviewer rr1J (Q6)**, we have added pseudocode in Appendix B.5 that illustrates the pipeline.

**To Reviewer zL5Z (Q1)**, we have added error bars in Table 1.

**To Reviewer zL5Z (Q2)**, we have compared the compute resources in Table 4.

**To Reviewer zL5Z (Q3) and (Q4)**, we have explicitly motivated the assumptions in Section 3.3.

**To Reviewer zL5Z (Q5)**, we have added a synthetic experimental validation in Appendix E.1.

**To Reviewer zL5Z (Q6)**, we have added an upper bound training by QA pairs generated by Qwen 2.5-VL-32B in Appendix.

**To Reviewer 7zkJ (Q2)**, we have added MMMU and MMMU-Pro in Table 1.

**To Reviewer 7zkJ (Q3)**, we provided an explanation for the backdoor adjustment.

Thanks again to the AC and reviewers for the time dedicated to reviewing this paper.

Sincerely,

The Authors

---

### Decision · Program_Chairs · 2025-09-17

**Decision:**

Accept (poster)

**Comment:**

This paper introduces a triangular consistency self-refinement framework for vision–language models (VLMs), demonstrating that models can autonomously improve using unlabeled images without external supervision. The idea is original and practically relevant, with clear causal-theoretical grounding and consistent empirical improvements across multiple benchmarks.

Pros:
* Novel principle (triangular consistency) to activate self-refinement in VLMs.
* Comprehensive experiments, including on LLaVA-1.5 (7B, 13B), MobileVLM, and Qwen-2.5-VL-3B.
* Ablation studies clarify that gains stem from consistency filtering rather than dataset size.
* Rebuttal and final revisions added error bars, training cost analysis, and societal impact discussion, addressing reviewer concerns.
* Reviewers agree that clarifications and additional experiments sufficiently resolved initial weaknesses.

Cons:
* Improvements, while consistent, are modest in scale.
* Training overhead (≈40 extra GPU hours) may limit practicality for some practitioners.
* Theoretical analysis relies on assumptions that remain somewhat abstract and hard to verify empirically.

Although the gains are incremental, the contribution is conceptually strong, methodologically sound, and well-supported by additional experiments. The reviewers’ concerns were adequately addressed, leading to raised or maintained positive scores. Overall, this is a solid and timely contribution to the study of self-refinement in VLMs and merits acceptance.